# Cadherins regulate nuclear topography and function of developing ocular motor circuitry

**Athene Knüfer[1], Giovanni Diana[1], Gregory S Walsh[2], Jonathan DW Clarke[1†*], Sarah Guthrie[3†*]**

[1]Centre for Developmental Neurobiology, King's College London, London, United Kingdom; [2]Department of Biology, Virginia Commonwealth University, Richmond, United States; [3]School of Life Sciences, University of Sussex, Brighton, United Kingdom

**Abstract** In the vertebrate central nervous system, groups of functionally related neurons, including cranial motor neurons of the brainstem, are frequently organised as nuclei. The molecular mechanisms governing the emergence of nuclear topography and circuit function are poorly understood. Here we investigate the role of cadherin-mediated adhesion in the development of zebrafish ocular motor (sub)nuclei. We find that developing ocular motor (sub)nuclei differentially express classical cadherins. Perturbing cadherin function in these neurons results in distinct defects in neuronal positioning, including scattering of dorsal cells and defective contralateral migration of ventral subnuclei. In addition, we show that cadherin-mediated interactions between adjacent subnuclei are critical for subnucleus position. We also find that disrupting cadherin adhesivity in dorsal oculomotor neurons impairs the larval optokinetic reflex, suggesting that neuronal clustering is important for co-ordinating circuit function. Our findings reveal that cadherins regulate distinct aspects of cranial motor neuron positioning and establish subnuclear topography and motor function.

**\*For correspondence:**
jon.clarke@kcl.ac.uk (JDWC);
S.Guthrie@sussex.ac.uk (SG)

†These authors contributed equally to this work

**Competing interests:** The authors declare that no competing interests exist.

## Introduction

During vertebrate embryonic development, neurons within the CNS assemble into stereotyped anatomical structures, such as clusters or layers (laminae). Neurons which share common inputs and axonal outputs, including motor neurons of the brainstem and spinal cord, are often organised as coherent clusters of cells, termed nuclei. Despite their prevalence, little is known about the mechanisms regulating their development (nucleogenesis). The correct arrangement of motor neurons within nuclei is thought to be critical for multiple aspects of circuit formation, including muscle target innervation and acquisition of sensory inputs (*Landmesser, 1978*; *Sürmeli et al., 2011*). Consequently, understanding the molecular programmes that direct motor neuron positioning and assembly of functional neuromuscular circuits represents an important challenge.

In the brainstem, some cranial motor nuclei, such as the oculomotor nucleus and facial motor nucleus, can be further divided into subnuclei according to their efferent projections. Neurons that contribute axons to the same cranial nerve yet innervate different muscle targets exhibit stereotyped topography within the larger nucleus (*Greaney et al., 2017*; *McArthur and Fetcho, 2017*; *Barsh et al., 2017*; *Isabella et al., 2020*). To date, the molecular and cellular processes which give rise to subnuclear topography remain unknown. The positioning of some groups of motor neurons has been shown to be regulated by members of the classical cadherin superfamily of cell adhesion molecules, which include type I and type II cadherins (*Price et al., 2002*; *Demireva et al., 2011*; *Astick et al., 2014*; *Dewitz et al., 2018*; *Dewitz et al., 2019*). Differential expression of type II

cadherins, which participate in homophilic as well as heterophilic interactions (*Patel et al., 2006*), drives the segregation and coalescence of somatomotor and branchiomotor neurons into discrete cranial motor nuclei in the hindbrain (*Astick et al., 2014*). Furthermore, cadherins participate in an intricate network of interactions with gap junctions and spontaneous activity, to regulate the emergence of mature topography in brainstem motor nuclei (*Montague et al., 2017*). Although the role of cadherins in the segregation of different motor neuron subtypes has been investigated, their requirement for the development of subnuclear architecture or motor function is unclear.

The zebrafish ocular motor system, which comprises the oculomotor nucleus (nIII) of the midbrain, as well as the trochlear (nIV) and abducens nuclei (nVI) of the hindbrain, provides an unparalleled opportunity to address these questions. The anatomical layout of the system is well-conserved among vertebrates; together the three nuclei innervate six extraocular muscles responsible for coordinating important eye movements (*Graf and McGurk, 1985*; *Evinger, 1988*; *Gilland and Baker, 2005*). Four of these muscles are innervated by the four constituent subnuclei of nIII, whereas nIV and nVI each target a single muscle. Oculomotor subnuclei are known to exhibit spatial topography with respect to muscle target, segregated along the dorsoventral axis (*Greaney et al., 2017*); in higher vertebrates, this organisation also extends to the visceral motor neurons of the Edinger-Westphal nucleus which is located dorsal to nIII (*Hasan et al., 2010*). Curiously, neurons belonging to the superior rectus (SR) subnucleus translocate their soma across the midline in an unusual migration to the contralateral side (*Puelles-Lopez et al., 1975*; *Naujoks-Manteuffel et al., 1991*), giving rise to contralateral connectivity.

In this study, we utilise the fine-grained anatomy and precise motor behaviour mediated by the zebrafish ocular motor system to identify the contribution of cadherin-mediated adhesion to the emergence of subnuclear topography and circuit function in vivo. We find that nIII subnuclei differentially express repertoires of type I and type II cadherins. Cranial motor neuron-specific perturbation of cadherin function interferes with normal programmes of neuronal positioning specific to each subnucleus, such as the clustering of dorsal oculomotor neurons and contralateral migration of ventral SR neurons. We also find that neurons belonging to the adjacent IO subnucleus, in which cadherin function is unperturbed, are mispositioned, suggesting that cadherin-mediated interactions between subnuclei contribute to their normal positioning. Finally, we find that cadherin function in dorsal oculomotor neurons is required for the development of efficient eye movements, as perturbing cadherin adhesivity impairs the nasal eye movements of the oculomotor-driven optokinetic reflex (OKR) predictably. Our findings show that cadherins play a fundamental role in the assembly of cranial motor subnuclei and circuitry and may facilitate the integration of motor neurons into functional circuits by controlling their stereotyped positioning.

## Results

### Developing ocular motor neurons differentially express cadherin combinations

In addition to their established roles in controlling motor neuron topography (*Price et al., 2002*; *Astick et al., 2014*), recent work indicates that type II cadherins act synergistically with the type I cadherin cdh2 (N-Cadherin) to control their precise positioning, although the exact mechanism remains unclear (*Dewitz et al., 2019*). In order to determine whether type I and type II cadherins are plausible candidates to play a role in ocular motor neuron development, we assayed the expression of several cadherins in developing ocular motor neurons using RNAscope fluorescent in situ hybridisation. Cadherin transcript detection was performed in transgenic *Isl1:GFP* zebrafish embryos, in which the majority of ocular motor neurons, including trochlear (nIV) and most oculomotor (nIII) neurons are known to express GFP (*Higashijima et al., 2000*). nIII neurons innervating the inferior oblique (IO) muscle, as well as abducens (nVI) motor neurons, lack GFP expression; we therefore focussed our analyses on nIII and nIV neurons which are GFP-positive.

*Isl1:GFP* fluorescence was preserved following RNAscope processing, allowing expression of individual mRNAs to be examined in ocular motor neurons, which display a fanned arrangement during development, within the intact brain (*Figure 1A*). Due to the lack of subpopulation-specific markers, expression in nIII subnuclei was assayed based on their anatomical location, as neurons are generated and distributed along the dorsoventral (DV) axis stereotypically (*Greaney et al., 2017*;

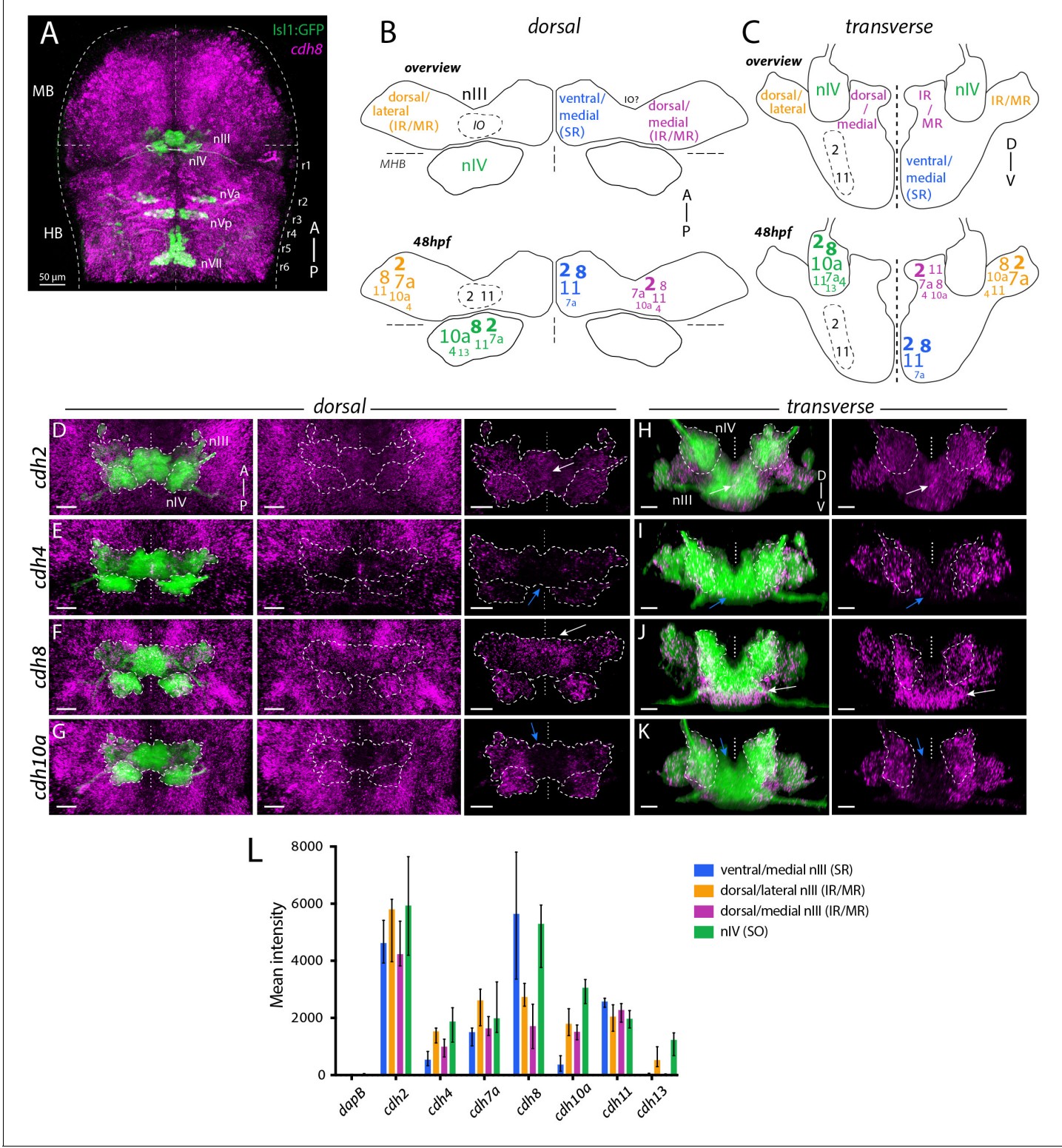

**Figure 1.** Differential expression of cadherin combinations in ocular motor neurons in *Isl1:GFP* zebrafish at 48 hpf. (**A**) Dorsal view of cranial motor neurons (green) and *cdh8* expression (magenta). Abbreviations: MB, midbrain; HB, hindbrain; r1-8, rhombomeres 1–8; nIII, oculomotor nucleus; nIV, trochlear nucleus; nVa, anterior trigeminal nucleus; nVp, posterior trigeminal nucleus; nVII, facial motor nucleus. Dotted lines indicate MHB, midline and outlines of the neural tube. (**B–C**) Combinatorial cadherin expression in nIII and nIV neurons. Font sizes indicate varying expression levels according to the median value of mean fluorescence intensities (in arbitrary units, a.u.) displayed in (**D**): small (500–1500 a.u.), regular (1500–2500 a.u.), large (2500–3500 a.u.) and large and bold font size (3500+ a.u.). Mean intensities below 500 a.u are omitted. Views represent flattened dorsal and transverse

*Figure 1 continued on next page*

*Figure 1 continued*

projections. IR, inferior rectus; MR, medial rectus; SR, superior rectus; IO, inferior oblique. See also *Figure 1—figure supplements 1* and *3*. (D–K) Expression of type I cadherins *cdh2* and *cdh4* and type II cadherins *cdh8* and *cdh10a* (magenta) in ocular motor neurons (green). White and blue arrows indicate high and low expression, outlines indicate location of nIII and nIV. All images are snapshots of 3D Imaris visualisations of entire z-stacks. Scale bars = 20 μm. (D–G) Dorsal views of ocular motor neurons and cadherin expression in surrounding tissue and masked by GFP-positive area. (H–K) Transverse views of ocular motor neurons and masked cadherin expression by GFP-positive area. (L) Corrected mean fluorescence intensities of cadherin expression in nIII and nIV neurons in a.u., detected using RNAscope in situ hybridisation. Data represents median and interquartile range, n = 5 measurements from n = 3 embryos, from three different clutches.

The online version of this article includes the following source data and figure supplement(s) for figure 1:

**Source data 1.**
**Figure supplement 1.** Differential expression of cadherins in ocular motor neurons in *Isl1:GFP* at 48 hpf.
**Figure supplement 2.** Detection of *foxg1a* mRNA expression at 48 hpf.
**Figure supplement 3.** Differential expression of cadherin combinations in ocular motor neurons in *Isl1:GFP* at 72 hpf.
**Figure supplement 3—source data 1.**

*Figure 1B–C*). Whereas the dorsal portion of nIII contains motor neuron groups innervating the inferior rectus (IR) and medial rectus (MR) muscles, ventral (and medial) nIII predominantly contains superior rectus (SR)-innervating neurons, as well as inferior oblique (IO) neurons just lateral to these (*Greaney et al., 2017*). Our data indicate that IO neurons are also found further dorsal, consistent with reports in other teleost species (*Graf and McGurk, 1985*; *Luiten and de Vlieger, 1978*). Superior oblique (SO)-innervating neurons belonging to nIV are located proximal to and just rostral of nIII in rhombomere 1 (r1) of the hindbrain.

Expression of mRNAs was surveyed at 48 hpf, ~2 hr before neurogenesis is complete in both nuclei (*Greaney et al., 2017*), and 72 hpf, when all extraocular motor neuron branches have innervated their respective muscle targets (*Clark et al., 2013*). To evaluate cadherin expression, we quantified average fluorescence intensities for each probe within nIV (SO-innervating) and three defined anatomical regions of nIII: a ventral medial region (SR-innervating), and a dorsal lateral and a dorsal medial region (both IR/MR-innervating). At 48 hpf, ocular motor neurons expressed combinations of cadherins (*Figure 1B–L*; *Figure 1—figure supplement 1B–D and F–G*), and differential expression was observed between dorsal and ventral nIII subnuclei, as well as between nIII and nIV. Detection was highly specific, as in situ hybridisation with a negative control probe against the bacterial mRNA *dapB* revealed almost no signal (*Figure 1—figure supplement 1A and E*), whereas expression of the transcription factor *foxg1a* within the CNS was restricted to telencephalic regions consistent with its known expression pattern (*Figure 1—figure supplement 2*; *Danesin et al., 2009*). Previous work has shown that selectivity of adhesion between cell types can be achieved by discrete profiles of cadherin expression, as well as by varying levels of the same cadherin (*Price et al., 2002*; *Steinberg and Takeichi, 1994*). We find that cadherin expression within ocular motor (sub) nuclei and surrounding non-motor neuron cell types varies both in intensity and spatial distribution. For example, expression of the type I cadherin *cdh2* (N-Cadherin) and type II cadherin *cdh11* was detected in all ocular motor neurons, displaying a broad and relatively uniform distribution across subnuclei, as well as cells immediately surrounding them, with *cdh2* displaying the highest levels of expression of all surveyed cadherins (*Figure 1D, H and L*; *Figure 1—figure*

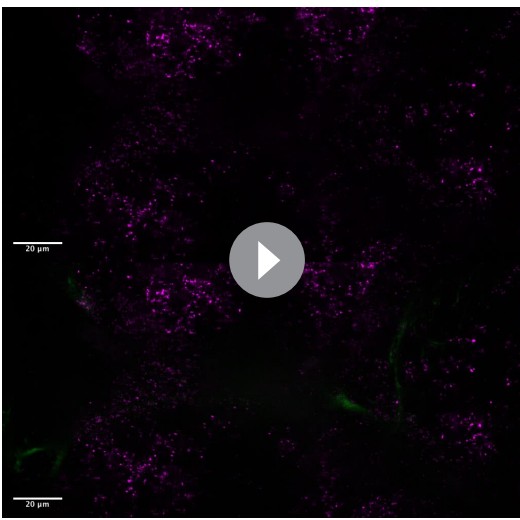

**Video 1.** Expression of *cdh8* (magenta) in ocular motor neurons (green) in *Tg(Isl1:GFP)* embryo at 48 hpf in confocal z-stack. Video displays z-stack moving from most ventral to dorsal and reveals high ventral expression.
https://elifesciences.org/articles/56725#video1

*supplement 1C and G*). Based on anatomical location, GFP-negative IO neurons are therefore also likely to express *cdh2* and *cdh11*, in addition to GFP-expressing ocular motor neurons.

Other cadherins exhibited more restricted expression patterns, with the most marked differences observed between dorsal and ventral nIII. Dorsal subnuclei expressed several cadherins, including *cdh4* (type I; *Figure 1E, I and L*) and *cdh10a* (type II; *Figure 1G, K and L*), which exhibited negligible expression in ventromedial nIII. In contrast, other cadherins were robustly expressed by all nIII neurons, but to varying degrees: expression of *cdh8* (type II; *Figure 1F, J and L*; *Video 1*) was far higher in ventral than in dorsal nIII, whereas *cdh7a* expression (type II; *Figure 1—figure supplement 1B, F and L*; *Video 2*) was lower in ventral neurons compared to dorsolateral nIII neurons. In addition, we observed differences between dorsolateral and dorsomedial domains for some cadherins, including *cdh7a* and *cdh8*, although these tended to be smaller than those observed between dorsal and ventral domains (*Figure 1L*). At 72 hpf, ocular motor neurons continued to express cadherin combinations (*Figure 1—figure supplement 3*), with continued broad CNS expression of *cdh11* and *cdh2*; *cdh2* continued to be the most highly expressed cadherin across both nuclei (*Figure 1—figure supplement 3C, J and Q*). Several cadherins showed higher overall expression in nIII compared to 48 hpf, most notably *cdh4* (*Figure 1—figure supplement 3D, K and Q*) and *cdh13* (*Figure 1—figure supplement 3I, P and Q*). Despite this, differential expression between dorsal and ventral nIII was maintained: expression of *cdh10a* remained absent from the most medial part of ventromedial nIII, resulting in lower levels of expression compared to dorsolateral nIII (*Figure 1—figure supplement 3G, N and Q*), whereas expression of *cdh8* was higher compared to dorsal nIII (*Figure 1—figure supplement 3F, M and Q*). IO neurons may also differentially express additional type II cadherins which display less uniform expression, although their lack of *Isl1:GFP* expression prevents us from evaluating this definitively. In sum, 2/7 cadherins surveyed showed general expression within ocular motor nuclei at both developmental stages, whereas the remaining cadherins displayed differential and dynamic expression across (sub)nuclei. These results reveal a complex organisation of cadherin expression in developing ocular motor neurons, whereby dorsal and ventral ocular motor subnuclei are defined by unique combinations of cadherin expression during the timeframe of nucleogenesis, and point to a role for both type I and type II cadherins in their development.

## Perturbing cadherin function disrupts clustering of dorsal oculomotor neurons

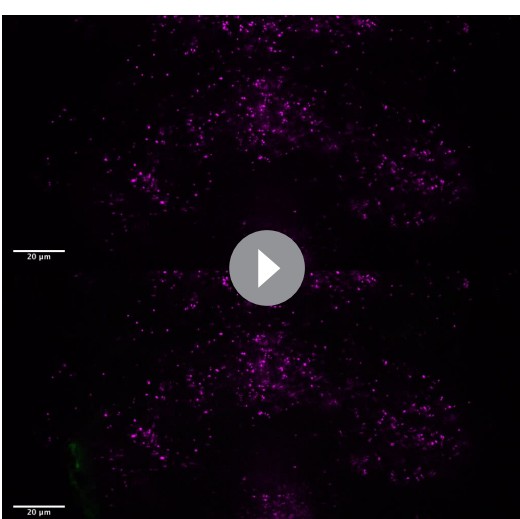

**Video 2.** Expression of *cdh7a* (magenta) in ocular motor neurons (green) in *Tg(Isl1:GFP)* embryo at 48 hpf in confocal z-stack. Video displays z-stack moving from most ventral to dorsal and reveals low ventral expression.

https://elifesciences.org/articles/56725#video2

As ocular motor neurons express a number of cadherins, we next sought to assay the role of cadherin expression during nucleogenesis by perturbing general cadherin adhesive function in cranial motor neurons in a stable transgenic line, Tg(*isl1:cdh2ΔEC-mCherry*)/Tg(*Isl1:GFP*), hereafter referred to as *cdh2ΔEC*. In this line, a dominant negative cadherin construct is expressed under the control of a cranial motor neuron-specific regulatory enhancer of the *Isl1* gene (*Rebman et al., 2016*). This construct (*cdh2ΔEC*) carries a large deletion encompassing ectodomains 1–4 of *cdh2*, and blocks cadherin adhesive function in an isotype-nonspecific manner, causing non-adhesive phenotypes in cells in which it is overexpressed by occupying sites where endogenous cadherins would otherwise localise, as well as by facilitating their endocytosis (*Rebman et al., 2016*; *Kintner, 1992*; *Nieman et al., 1999*; *Jontes et al., 2004*). As a result, it is expected to perturb both type I and type II cadherin-mediated adhesion.

*Figure 2* In order to examine the effect of perturbing cadherin function on the organisation of ocular motor neurons, we performed live

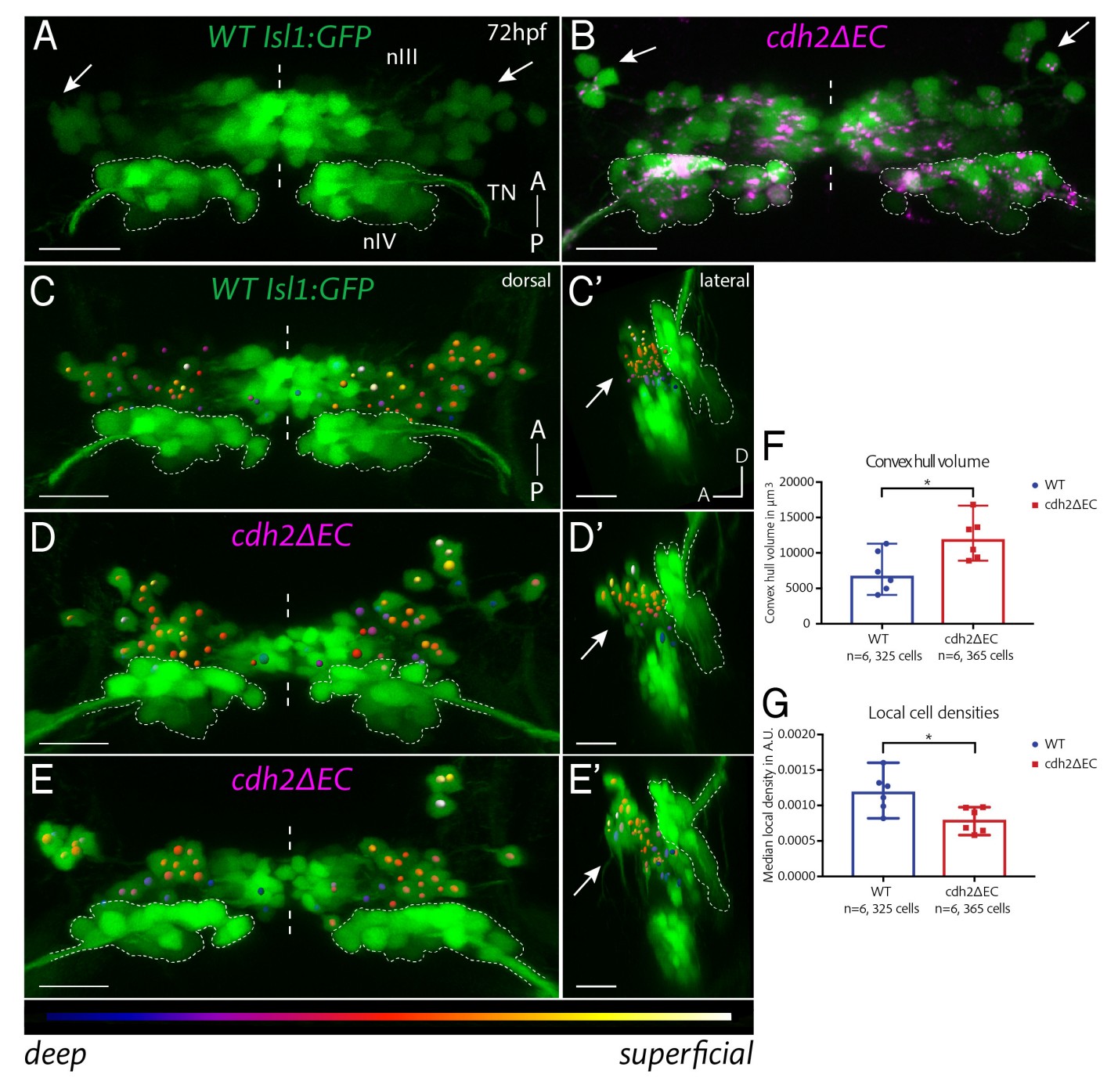

**Figure 2.** Expression of *cdh2ΔEC* in zebrafish ocular motor neurons causes scattering of dorsal oculomotor neurons. (A–B) Dorsal maximum intensity projections of nIII and nIV (green) in wild-type (A) *Isl1:GFP* and *cdh2ΔEC* larvae (B) at 72 hpf. Arrows point to dorsal neurons which lie in lateral regions of nIII. TN, trochlear nerve. Scale bars = 20 μm. See also *Figure 2—figure supplement 1*. (C–E') Oculomotor coordinates in dorsal half of nIII shown in dorsal and lateral views of wild-type *Isl1:GFP* and *cdh2ΔEC* larvae at 72 hpf, colour coded according to z-level. Arrows point to dorsal nIII neurons which are mispositioned in *cdh2ΔEC* fish. All images are snapshots of 3D Imaris visualisations of entire z-stacks. Outlines indicate location of nIV. Scale bars = 20 μm. (F) Convex hull volumes of dorsal cell coordinates in μ$^3$. Data represents median and range, n = 6 larvae from two clutches for each genotype, *p=0.0411, two-tailed Mann-Whitney U. (G) Local cell densities derived from dorsal cell coordinates in arbitrary units. Data represents median and range, n = 6 larvae from two clutches for each genotype, *p=0.0152. There was no significant difference in the number of cells measured per side (medians of 28.25 for wild types and 29.75 for *cdh2ΔEC*; n = 6 larvae per genotype, p=0.3074). Both tests are two-tailed Mann-Whitney U. The online version of this article includes the following source data and figure supplement(s) for figure 2:

*Figure 2 continued on next page*

*Figure 2 continued*

**Source data 1.**

**Figure supplement 1.** Ocular motor nuclei of *cdh2ΔEC* larvae retain key characteristics of dorsoventral organisation but appear disorganised compared to wild types.

imaging of wild-type *Isl1:GFP* and *cdh2ΔEC* larvae at 72 hpf (*Figure 2*), when both nIII and nIV are complete in terms of absolute numbers of terminally differentiated neurons and have also innervated their appropriate muscle targets during normal development. Live imaging revealed expression of *cdh2ΔEC*, as evidenced by punctate mCherry-positive fluorescence, in all GFP-positive nIII neurons, however no signal was observed in adjacent GFP-negative IO neurons, nVI neurons or other surrounding cell types, recapitulating the lack of expression observed in *Isl1:GFP* (*Figure 2B*).

Following perturbation of cadherin function, dorsal and ventral nIII populations were still visibly distinct as in wild types (*Figure 2—figure supplement 1*), however the positioning of dorsal nIII neurons was perturbed (*Figure 2B*). Two main phenotypes were observed: Firstly, neurons were more sparsely distributed than their wild-type counterparts (*Figure 2C–D'*). Secondly, they were often also scattered over a wider area, so that occasionally small groups of dorsal cells were found clustered a greater distance away from the rest of the nucleus (*Figure 2E–E'*). Interestingly, the density and positioning of neurons located in ventromedial nIII was not perturbed. In order to measure the degree of dispersion, as well as the overall spatial extent of dorsal nIII neurons, three-dimensional cell coordinates were recorded for the dorsal half of nIII (*Figure 2C–E'*). The number of cells measured per side, per larva did not differ depending on the genotype (*Figure 2G*), suggesting that normal programmes of differentiation were unlikely to be affected following perturbation of cadherin function, recapitulating previous findings (*Astick et al., 2014*). However, there was a significant increase in the convex hull volumes (the volume of the smallest convex set in Euclidean space containing all coordinates) of dorsal nIII neurons in *cdh2ΔEC* larvae (*Figure 2F*), demonstrating that these neurons occupy a larger area in 3D space as a result of disrupted cadherin function. Furthermore, local cell densities estimated by Delaunay tessellation were significantly reduced in these larvae compared to wild types (*Figure 2G*), indicating that, overall, *cdh2ΔEC*-expressing neurons in dorsal nIII have fewer oculomotor neighbours as a result of weakened adhesions. We conclude that cadherins are required for the clustering and correct positioning of IR/MR subnuclei of nIII.

## Cadherins are required for contralateral migration of ventral superior rectus neurons

In the developing CNS, contralateral connectivity is predominantly established via midline-crossing commissural axons, which are guided by axon guidance factors secreted by midline cells (*Ypsilanti et al., 2010*). In contrast, the contralateral innervation pattern of superior rectus (SR) neurons, which are located in ventral nIII, is the product of an unusual and well-conserved migration during which they translocate their soma across the midline to the contralateral side (*Figure 3A*). Studies in other vertebrates have shown that SR neurons send out axons ipsilaterally and migrate to the contralateral side along a leading process which extends into the contralateral nucleus, resulting in contralateral projection (*Puelles-Lopez et al., 1975*; *Bjorke et al., 2016*; *Dun et al., 2012*).

In zebrafish, SR neurons which have completed this migration can be detected from 5 dpf (*Greaney et al., 2017*); however, their development has not been fully described in this organism. We initially sought to characterise the time frame of zebrafish SR contralateral migration in early development. Using live time-lapse imaging of wild-type *Isl1:GFP* embryos, we found that bilateral ventral nIII neurons are initially separated at ~43.5 hpf and progressively converge towards the midline, until cells of either side are in contact from ~47 hpf. Using immunohistochemistry against GFP in order to examine their arrangement, we found that ventral nIII neurons are found interdigitated at the midline by 72 hpf, suggesting that migration is underway (*Figure 3B*). We estimate that SR neuron migration is complete by ~4 dpf based on contralateral cells counts following unilateral application of DiI to the orbit of fixed *Isl1:GFP* embryos of various developmental stages, (data not shown).

We next asked whether cadherin-adhesive function could play a role in regulating SR neuron migration by examining the organisation of ventral nIII neurons at the midline at 50 hpf, when midline convergence is usually complete. Whereas wild-type ventral nIII neurons were found fully

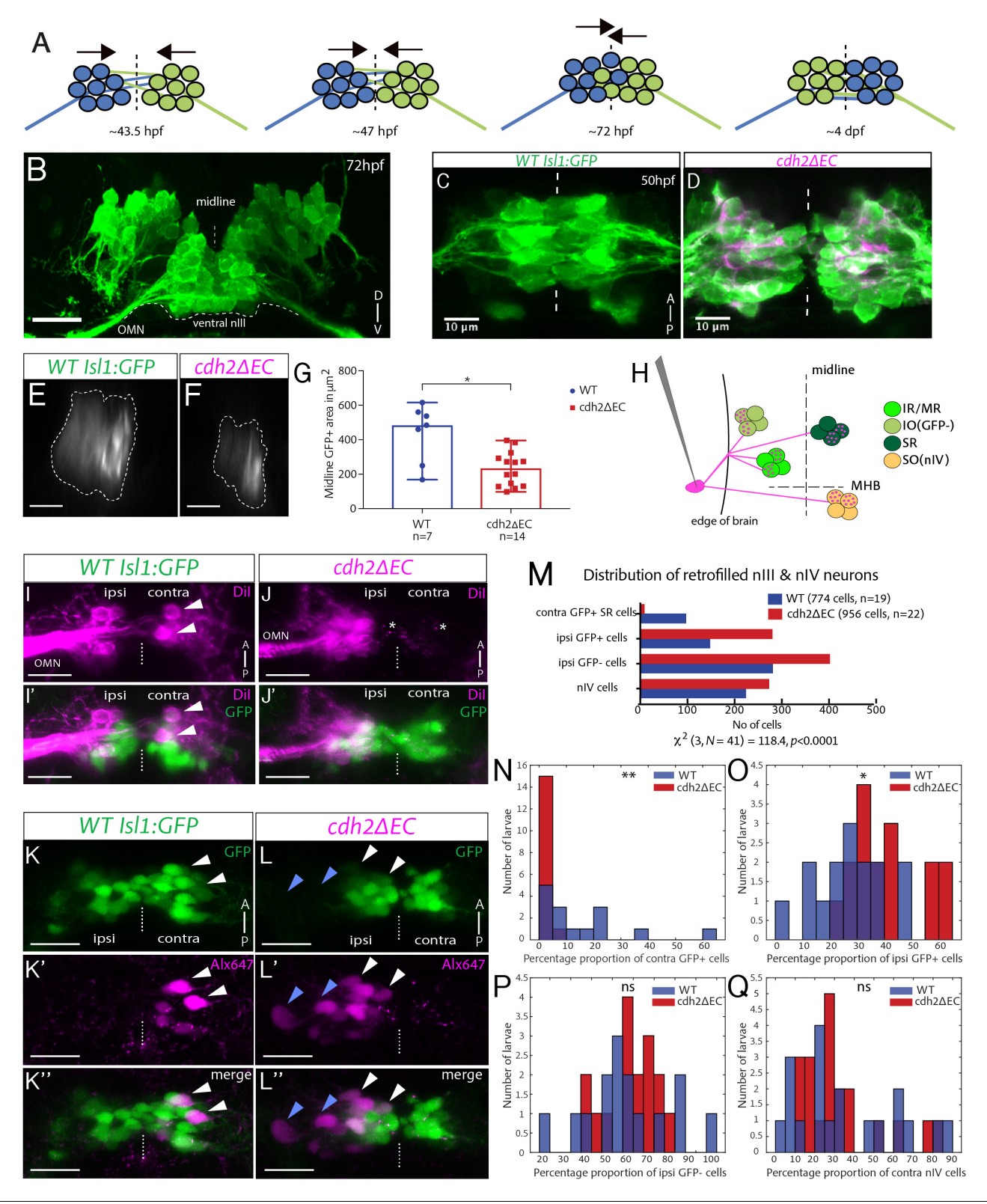

**Figure 3.** Contralateral migration of ventral SR neurons is abolished in *cdh2ΔEC* fish. (**A**) Dorsal view schematic of contralateral migration phases. Dotted lines and arrows indicate the midline and direction of migration, lines represent leading processes. (**B**) Transverse view projection of an *Isl1:GFP* embryo immunostained against GFP. TN, trochlear nerve. Scale bars = 20 µm. (**C–D**) Dorsal view maximum intensity projections of ventral half of wild-type *Isl1:GFP* and *cdh2ΔEC* embryos immunostained against GFP and RFP at 50 hpf. Dotted lines represent the midline. (**E–F**) Single sagittal confocal

*Figure 3 continued on next page*

*Figure 3 continued*

sections of nIII GFP fluorescence at the midline in wild-type and *cdh2ΔEC* fish. Scale bars = 20 µm. (**G**) Area of GFP fluorescence measured in sagittal sections of nIII. Error bars represent median and range, n = 7 (wild-type) from two clutches and n = 14 from two clutches (*cdh2ΔEC*), *p=0.0125. There was no difference in the number of nIII cells (medians for wild types (n = 4)=95 and *cdh2ΔEC* (n = 9)=103, p=0.414). Both tests are two-tailed Mann-Whitney U. (**H**) Labelling of ocular motor subnuclei by retro-orbital dye fill. MHB, midbrain-hindbrain boundary. (**I–J'**) Ventral view maximum intensity projections of DiI-filled ventral nIII neurons in *Isl1:GFP* and *cdh2ΔEC* larvae at 5 dpf. Scale bars = 20 µm. White arrows indicate neurons which have crossed to the contralateral side, asterisks represent mCherry+ signal. (**K–L''**) Dorsal view maximum intensity projections of sparsely labelled neurons located in ventral nIII in *Isl1:GFP* and *cdh2ΔEC* larvae at 5 dpf. White and blue arrows indicate GFP-positive and GFP-negative dye-filled neurons. The left eye is the side of dye fill. Alx647 = AlexaFluor 647 dye. Scale bars = 20 µm. Note that in K-K'', small magenta puncta represent non-specific dye spread, while in K-K'' these likely also include punctate mCherry signal from *cdh2ΔEC* expression, due to spectral overlap. (**M**) Total observed number of pooled dye-filled cells belonging to each category in wild types and *cdh2ΔEC* at 5 dpf. There is a significant relationship between genotype and distribution of dye-filled neurons, (p<0.0001; chi squared test). There was no significant effect of genotype on the total number of dye-filled neurons across groups per larva, medians for wild types = 31, *cdh2ΔEC* = 40.5, p=0.674, 2-tailed Mann-Whitney U. Wild-type larvae are from four clutches and *cdh2ΔEC* larvae are from five clutches. (**N–Q**) Percentage proportions of dye-filled neurons per larva for each genotype separated by category. All tests are two-tailed Mann-Whitney U. (**N**) Percentage proportions of contralateral GFP-positive SR cells. n = 15 larvae (wild-type) and n = 16 (*cdh2ΔEC*) larvae, **p=0.0015. (**O**) Percentage proportions of ipsilateral GFP+ cells. n = 15 larvae (wild-type) and n = 16 (*cdh2ΔEC*) larvae, *p=0.0109. (**P**) Percentage proportions of ipsilateral GFP negative IO cells. n = 15 larvae (wild-type) and n = 16 (*cdh2ΔEC*) larvae, p=0.5131. (**Q**) Percentage proportions of contralateral nIV cells. n = 19 larvae (wild-type) and n = 20 (*cdh2ΔEC*) larvae, p=0.9391. (**N–P**) Proportions are relative to the total number of nIII cells filled per larva, for (**Q**) proportions are relative to combined total of dye-filled nIII and nIV neurons. Larvae in which less than 10 nIII cells were filled in total were excluded from analysis in (**N–P**), whereas larvae in which less than 10 combined nIII and nIV cells were filled in total were excluded from analysis in (**Q**).

The online version of this article includes the following source data for figure 3:

**Source data 1.**

interdigitated at the midline in the majority of embryos (***Figure 3C***), only a few *cdh2ΔEC*-expressing neurons had successfully converged, leaving a gap between large numbers of cells (***Figure 3D***). In order to quantify the degree of midline convergence, we measured the area of GFP fluorescence present in a single sagittal section directly at the midline, obtained from resliced z-stacks (***Figure 3E–G***). This was always seen as one large area rather than multiple smaller spots (irrespective of genotype), indicating that cells move towards the midline in a collective fashion. There was a significant decrease in the area of GFP fluorescence imaged in *cdh2ΔEC* embryos (***Figure 3F and G***) compared to wild types (***Figure 3E***), whereas no difference was detected in the total numbers of nIII neurons between them. This suggests that the convergence of SR neurons towards the midline is perturbed in *cdh2ΔEC* embryos, which is unlikely to be due to delayed neurogenesis. Interestingly, our live imaging experiments at 72 hpf show that *cdh2ΔEC*-expressing SR neurons eventually reach the midline (***Figure 2B and D–E***). Overall, these results indicate that perturbing cadherin function delays, but does not prevent, the convergence of SR neurons towards the midline.

We next sought to investigate whether cadherin adhesivity regulates later stages of contralateral migration, during which SR neurons cross the midline and migrate into the contralateral nucleus. Using unilateral application of DiI to the oculomotor nerve of fixed larvae at 5 dpf, we labelled groups of contralateral oculomotor neurons in wild types (***Figure 3I–I'***); however, in *cdh2ΔEC* larvae, nIII-labelled neurons were only found on the ipsilateral side (***Figure 3J–J'***), suggesting that contralateral migration is abolished following the loss of cadherin adhesion. In order to examine the distribution of SR neurons in greater detail, we adopted a sparse labelling approach at the same developmental stage. Using unilateral retro-orbital dye fills, oculomotor neurons of *Isl1:GFP* larvae can be traced from their muscle targets and assigned a subnuclear identity based on presence (or absence) of GFP fluorescence and anatomical location (***Figure 3H***; ***Greaney et al., 2017***). Live imaging revealed dye-filled GFP-positive (SR) cells on the contralateral side in wild types (***Figure 3K–K''***), whereas there was a dramatic reduction in the number of contralateral cells labelled in *cdh2ΔEC* larvae, despite successful labelling of ipsilateral neurons (***Figure 3L–L''***). Examination of the pooled data revealed a significant relationship between genotype and subnuclear identity (***Figure 3M***). There was a significant reduction in the proportion of contralateral GFP-positive nIII cells in *cdh2ΔEC* larvae compared to wild types (***Figure 3N***), with 15/16 larvae exhibiting proportions of <5% of labelled SR neurons compared with 10/15 wild-type larvae, which had proportions of >5% ranging up to 65%. We reasoned that there should be a concomitant increase in the proportions of dye-filled ipsilateral GFP-positive cells, should *cdh2ΔEC* expression have interfered with the ability of SR

neurons to migrate. Indeed, there was a significant increase in the proportion of ipsilateral GFP-positive cells in *cdh2ΔEC* larvae (*Figure 3O*), which likely reflects the additional detection of SR neurons which have failed to migrate to the contralateral side, in addition to IR/MR cells. In contrast, we found no difference in the proportions of dye-filled IO (*Figure 3P*) or SO/nIV neurons between genotypes (*Figure 3Q*). Taken together, our results indicate that the contralateral migration of SR neurons depends on cadherin-mediated adhesion. Our finding that SR neurons with perturbed cadherin adhesivity eventually reach the midline, but do not cross it, suggests that contralateral migration proceeds via a two-phase process: an initial convergence towards the midline, followed by somal translocation across it.

### Dominant negative cadherin expression results in mispositioning of trochlear neurons along the dorsoventral axis

In addition to nIII, the trochlear nucleus (nIV), which is located proximal to nIII in r1 of the hindbrain is labelled in *Isl1:GFP* fish. To investigate the effect of cadherin perturbation on the positioning of nIV neurons, we performed live imaging at 72 hpf using lateral mounting. Interestingly, although nIV neurons were not scattered, they were aberrantly positioned along the DV axis, resulting in a significantly dorsally elongated trochlear nucleus compared to wild-type fish (*Figure 4*). The position of nIV did not appear to be shifted overall with respect to nIII. During wild-type development, nIV elongates along the DV axis between 48hpf and 72hpf (unpublished data), suggesting the presence of guidance cues or mechanical forces which act along this axis. We speculate that abrogating cadherin function in these neurons impairs the cohesion of the nucleus, resulting in its increased elongation.

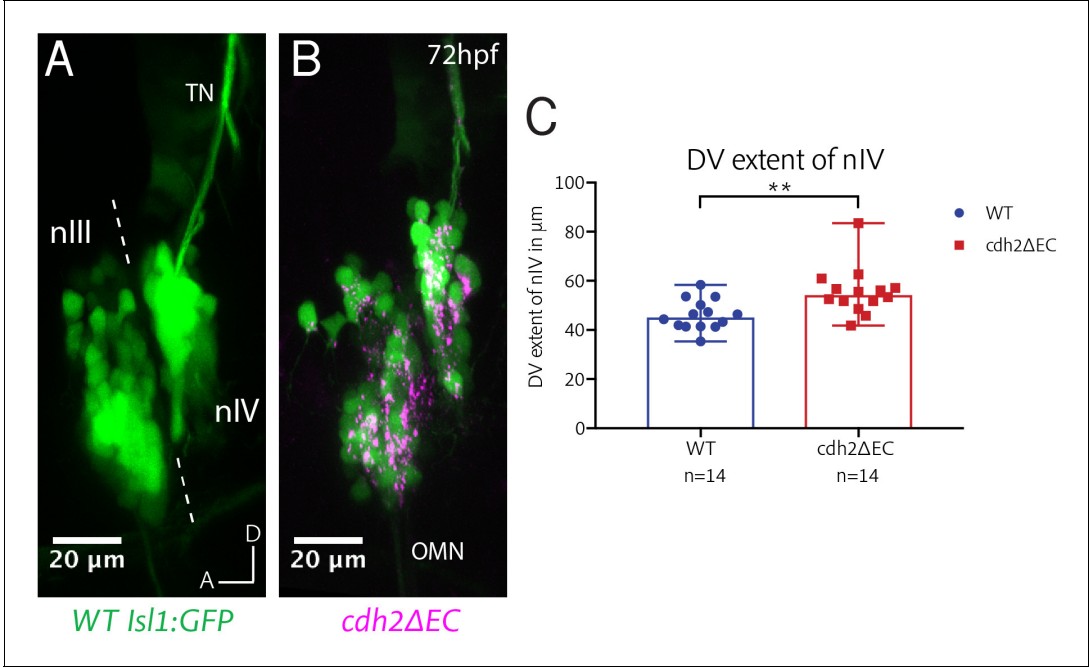

**Figure 4.** *cdh2ΔEC* expression results in mispositioning of zebrafish trochlear neurons along the dorsoventral axis. (A–B) Lateral view maximum intensity projections of nIII and nIV of wild-type *Isl1:GFP* and *cdh2ΔEC* larvae at 72 hpf. Dotted lines represent the midbrain-hindbrain boundary. OMN, oculomotor nerve; TN, trochlear nerve. (C) Measurements of dorsoventral extent of nIV at 72 hpf. Dots represent individual larvae. Error bars represent median and range. n = 14 larvae from two clutches for each genotype, **p=0.0042, two-tailed Mann-Whitney U. Larvae were mounted laterally to align both eyes prior to imaging. Where necessary, images were resampled to align bilateral ocular motor nuclei to the same plane and oriented so that the edge of the neuropil was directly parallel to the anterior-posterior axis.

The online version of this article includes the following source data for figure 4:

**Source data 1.**

## IO neurons require interactions with neighbouring nIII subpopulations for precise pool positioning

The above experiments identify subpopulation-specific requirements for cadherins in the correct positioning of ocular motor neurons. We next asked whether subnucleus position depends in part on cadherin-based interactions between neurons of different subnuclei, which are spatially proximal. In *cdh2ΔEC* embryos, neurons belonging to the inferior oblique (IO) nucleus were never seen to express *cdh2ΔEC-mCherry*, allowing us to ask whether their position is dependent on cadherin function in other subnuclei. Several cadherins, including *cdh2* and *cdh11*, are expressed by all oculomotor neurons, including IO neurons, whereas other type II cadherins which are capable of interacting heterophilically with *cdh11*, such as *cdh8*, are expressed differentially by subnuclei (*Figure 1B–C*; *Brasch et al., 2018*). This raises the possibility that homophilic and heterophilic cadherin-mediated adhesive interactions could arise between IO neurons and other oculomotor neurons. We sparsely labelled IO neurons using retro-orbital dye fills as above and analysed their distribution according to their dorsoventral (DV; 15 bins) and mediolateral (ML; four bins) location. We assigned a coordinate to each dye-filled neuron within an overall grid of 60 bins, corresponding to the full spatial extent of nIII. 3D-visualisation of nIII revealed that the IO subnucleus resembles a columnar organisation along the DV axis, with neurons located ventral and lateral to SR neurons, as well as dorsally and medially (*Figure 5A–A''*).

We first assessed differences in the distribution of IO neuron location using a multivariate extension of the Kolmogorov-Smirnov test. This revealed that the overall cellular distribution for IO neurons was significantly different between genotypes (k = 0.2427, p<0.0001). In wild types, probabilities of IO location are biased towards medial locations and are found along the DV axis, with slightly higher probabilities for dorsal regions (*Figure 5B*). In *cdh2ΔEC* larvae, reduced dorsal and slightly more lateral probabilities were observed for IO neurons (*Figure 5C–D*); however, the overall arrangement of IO neurons as a subnucleus appeared similar. In order to estimate the full extent of IO neuron distribution, we used a non-parametric Bayesian approach where the probability of a neuron to be placed in any of the 60 bins in the grid is described as a Dirichlet distribution, which is then estimated by performing statistical inference from the observed data. Estimated DV-ML occupancy distributions confirmed differences in the overall spatial distribution of IO neurons between genotypes (*Figure 5E–G*). We investigated the directionality of the shift along the two major axes and found an overall tendency towards higher, dorsal values for DV subdivision in wild types compared to *cdh2ΔEC* larvae (*Figure 5H*). Additionally, there was a slight tendency overall to lower, medial values for wild types (*Figure 5I*). This demonstrates that overall, the positioning of IO neurons of *cdh2ΔEC* larvae is altered compared to wild types: IO neurons are located more ventral and lateral, consistent with a shift away from adjacent IR/MR and SR subnuclei (*Figure 5J–K*). In sum, these results suggest that cadherin-dependent neuron-neuron adhesive interactions between oculomotor subnuclei also contribute to the appropriate positioning of the IO subnucleus. However, we cannot rule out an alternative possibility that the observed shift in IO subnucleus position could also be a mechanically driven consequence of changes to the arrangement of the other oculomotor subnuclei; novel tools to target expression of *cdh2ΔEC* to IO neurons specifically would allow us to address this possibility definitively.

## Neuromuscular connectivity is preserved in *cdh2ΔEC*-expressing ocular motor neurons

Whereas some studies report that perturbations to cadherin signalling in hindbrain cranial motor neurons affect the positioning of soma, while preserving stereotyped patterns of axon outgrowth and pathfinding (*Astick et al., 2014*; *Rebman et al., 2016*), others have identified a role for specific classical cadherins in the regulation of normal axonal development in motor neurons, including pathfinding, axon outgrowth and branching (*Brusés, 2011*; *Barnes et al., 2010*). To investigate the role of cadherin adhesive function in the development of peripheral ocular motor connectivity, we examined the efferent projections of nIII and nIV neurons in *cdh2ΔEC* larvae using confocal imaging from the lateral view. At 72 hpf, all ocular motor nerves and branches were present and appeared to have reached their respective EOMs (*Figure 6A–B*). In order to examine possible defasciculation defects, we quantified the number of axon branches belonging to each nerve bundle for each of the four oculomotor nerve (OMN) branches, as well as for the trochlear nerve (TN). There was no significant

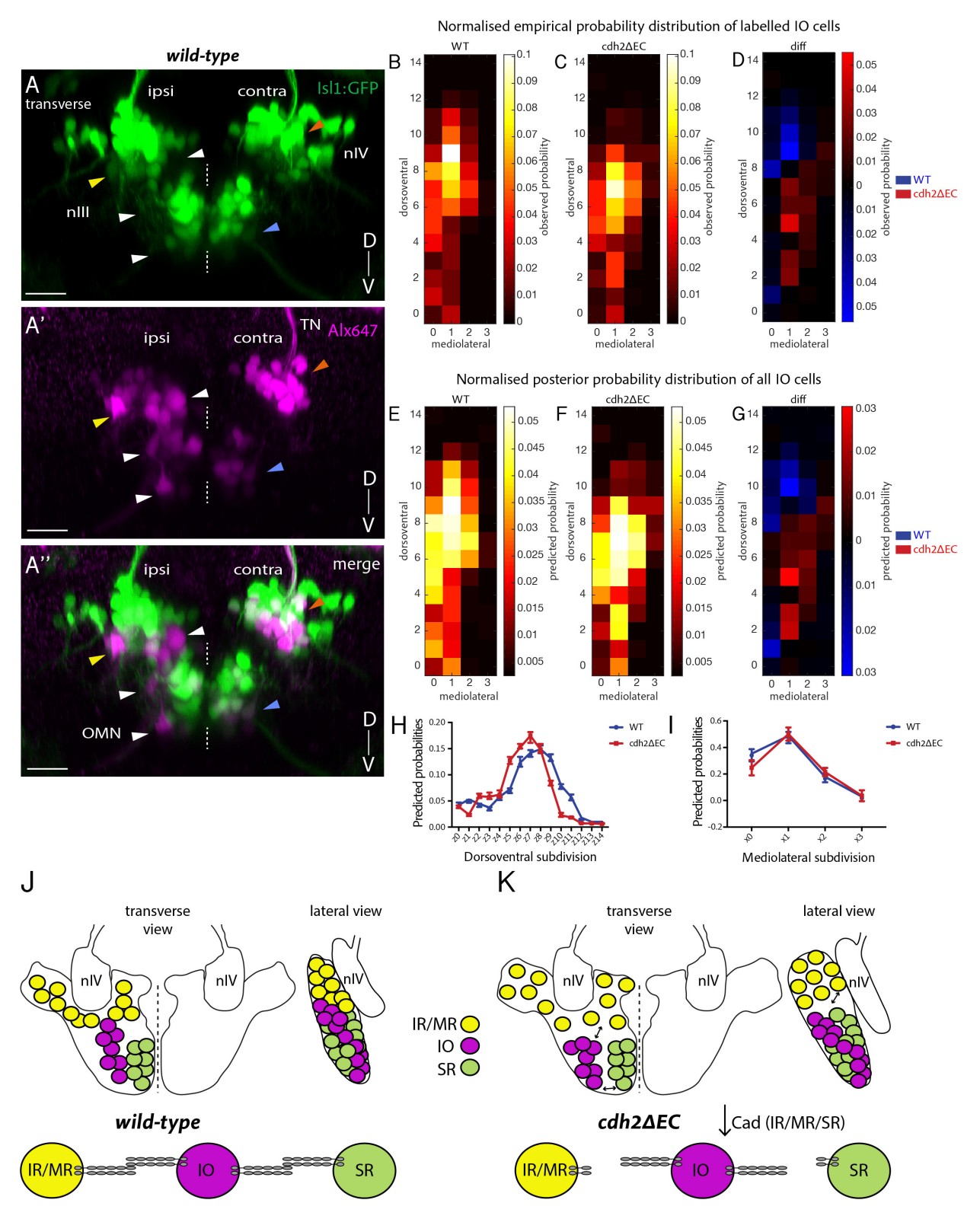

**Figure 5.** Non-*cdh2ΔEC* expressing zebrafish inferior oblique neurons are mispositioned following perturbation of cadherin function in neighbouring oculomotor subnuclei. (**A–A''**) Transverse maximum intensity projection of a wild-type *Isl1:GFP* larva at 5 dpf in which nIII and nIV neurons have been labelled using retro-orbital dye fills with Alexa647 dextrans. Ipsi denotes side of dye fill. White and yellow arrows indicate ipsilateral GFP-negative IO neurons and ipsilateral GFP-positive neurons (IR/MR). Blue and orange arrows indicate contralateral GFP-positive SR neurons and SO (nIV) neurons.

*Figure 5 continued on next page*

*Figure 5 continued*

OMN, oculomotor nerve; TN, trochlear nerve. Scale bars = 20 µm. (B–C) Normalised empirical and posterior (E–F) probability distribution of IO neuron spatial locations in wild types (278 cells, n = 19 larvae, four clutches) and *cdh2ΔEC* (399 cells, n = 22 larvae, five clutches) according to dorsoventral (DV; from z0 (ventral) to z14 (dorsal)) and mediolateral (ML; from x0 (medial) to x3 (lateral)) subdivisions. (D) Normalised empirical and posterior (G) differential probability distribution of IO neuron spatial locations between genotypes. diff = *cdh2ΔEC* distribution – wild-type distribution. There are differences in the overall spatial distribution between genotypes ($KL_{mean}$ = 0.3114, 95% CI [0.3107 0.3122]). (H–I) Median posterior probabilities of IO cell location within DV (H) and ML (I) subdivisions for wild types and *cdh2ΔEC* larvae. There is a difference in DV (*z-shift* = 0.5201, 95% CI [0.5168 0.5234]) and ML distribution (*x-shift* = −0.1423, 95% CI [−0.1431–0.1414]) between genotypes. (J–K) Diagrammatic representation of transverse and lateral views of IO ocular motor neuron location in wild types and *cdh2ΔEC*. Perturbation of cadherin binding between *cdh2ΔEC*-expressing IR/MR and SR subnuclei and wild-type IO neurons leads to a ventral and lateral shift in IO neuron location.

difference in the number of branches between genotypes (*Figure 6C*). These observations indicate that axon outgrowth, targeting and fasciculation of ocular motor axons is robust to cell-autonomous dominant negative cadherin expression.

Once ocular motor axons have navigated to their appropriate targets, initial contacts with the target muscle stabilise and form neuromuscular junctions (NMJs). We have previously shown that synaptogenesis is underway in all muscles from 72 hpf (*Clark et al., 2013*), correlating with the onset of spontaneous saccadic eye movements (*Easter and Nicola, 1997*). We next asked whether cadherin function could be required for the formation of NMJs. Cadherins have known roles in the formation of central synapses (reviewed in *Salinas and Price, 2005*) and cdh2 is known to localise at developing NMJs (*Cifuentes-Diaz et al., 1994*; *Brusés, 2011*). We performed immunohistochemistry against the presynaptic vesicle protein SV2 and labelling of postsynaptic acetylcholine receptors using fluorescently tagged α-bungarotoxin at 4 dpf and identified extraocular NMJs according to their spatial location using brightfield signal. We found that pre- and postsynaptic components colocalise as discrete patches on respective EOMs both in wild types and *cdh2ΔEC* larvae (*Figure 6—figure supplement 1A–H''*), including on the medial rectus (MR) muscle (*Figure 6D–G''*), which is one of two muscles driving horizontal eye movements of the larval optokinetic reflex. Furthermore, we observed no difference in the overall area of SV2 signal (*Figure 6—figure supplement 1I*) or degree of colocalisation (*Figure 6—figure supplement 1J*), suggesting that NMJs form properly following perturbation of cadherin adhesion. However, we cannot exclude an effect on synapse stabilisation at later stages of development or in the regulation of synaptic strength. Taken together, our findings suggest that perturbing cadherin function does not erode the peripheral connection between ocular motor neurons and muscle targets.

## Abrogating cadherin function in developing oculomotor neurons causes eye movement defects

We next considered whether perturbing cadherin-mediated adhesion in oculomotor neurons could impact their emergent motor function by assaying the larval optokinetic reflex (OKR). The OKR is a well-conserved gaze-stabilising reflex which consists of slow-phase and fast reset eye movements in response to optic flow. In zebrafish, the OKR is robust by 3–4 dpf (*Easter and Nicola, 1997*). We measured the OKR of immobilised zebrafish at 5 dpf by capturing the eye movements of larvae, which were placed within a circular arena, from above with a high-speed video camera as they viewed vertical black and white moving bars of varying speed, contrast, and spatial frequency. The horizontal eye movements of the OKR rely on the action of two antagonistic EOMs, the oculomotor-driven medial rectus and abducens nucleus-controlled lateral rectus (*Figure 7B*). These muscles drive nasal (adduction) and temporal (abduction) eye movements, respectively (*Figure 7A–A'*). As nVI neurons of *cdh2ΔEC* larvae do not express *cdh2ΔEC*, expression is restricted to MR neurons which are known to be contained within dorsal nIII and display a severely scattered phenotype in *cdh2ΔEC* larvae (*Figure 2*). We therefore hypothesised that cell-specific perturbation of cadherin function would primarily affect nasal rather than temporal eye movements.

We analysed movement for each individual eye and evalu ated the eye gain of the slow-tracking eye movement separately for nasal and temporal movements, as well as eye range. We consistently observed that larvae displayed a greater range of movement in one eye and therefore assigned each eye as either the strong or weak eye. We found that *cdh2ΔEC* larvae displayed severe defects in the efficiency of their nasal eye movements when compared to wild types (*Figure 7C*), exhibiting

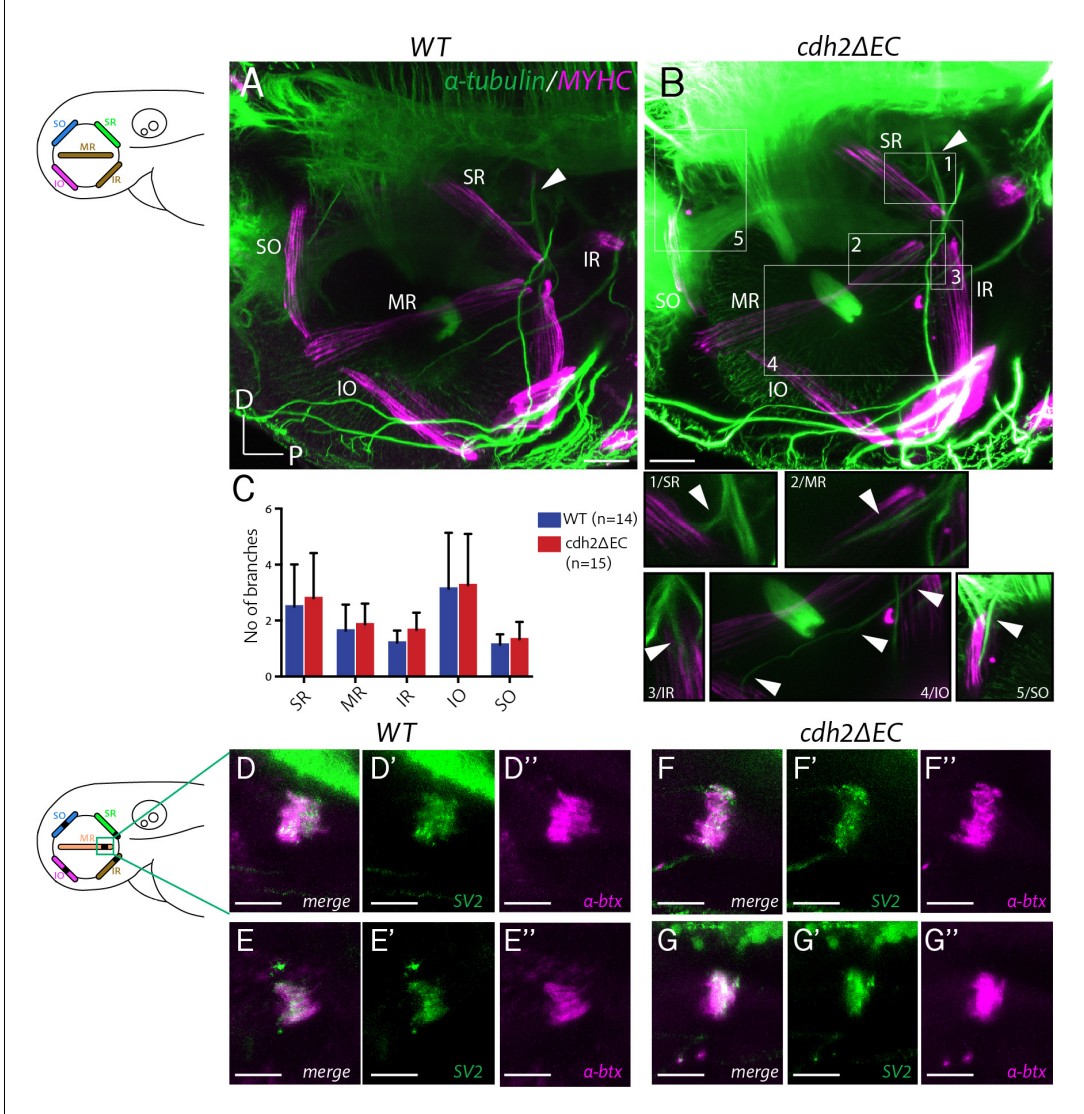

**Figure 6.** Extraocular muscle innervation and neuromuscular junction formation is unperturbed in *cdh2ΔEC* larvae. (**A–B**) Immunostained axon projections of oculomotor and trochlear nerves (α-tubulin) innervating extraocular muscles (myosin heavy chain) at 72 hpf in wild types and *cdh2ΔEC-mCherry*. Images are lateral view maximum intensity projections, white arrow indicates the point at which the oculomotor nerve enters the orbit. Boxes show high magnification view of individual branches. Scale bars = 40 μm. (**C**) Quantification of branch number per genotype for each nerve/branch. There was no significant difference in branch number for any muscle target (ns; SR, p=0.9406; MR, p=0.9451; IR, p=0.8512; IO, p=0.9451; SO, p=0.94512; multiple t-tests, and Holm-Sidak test). Error bars are mean and SD. Both wild-type and *cdh2ΔEC* larvae are from three clutches. (**D–G''**) Immunostained NMJs composed of presynaptic vesicles (SV2) and acetylcholine receptors (α-bungarotoxin) in wild types (**D–E''**) and *cdh2ΔEC* (**F–G''**) at 4 dpf at the medial rectus (MR) muscle. Scale bars = 20 μm. All images are maximum intensity projections of sub-stacks taken from larger z-stacks. See also *Figure 6—figure supplement 1*.

The online version of this article includes the following source data and figure supplement(s) for figure 6:

**Source data 1.**
**Figure supplement 1.** Perturbation of cadherin function does not disrupt zebrafish extraocular neuromuscular junction formation.
**Figure supplement 1—source data 1.**

a significant decrease in the eye gain for both eyes across all trials, with the strongest effect seen for the strong eye. Conversely, we found no difference in the eye gain for temporal eye movements between *cdh2ΔEC* larvae and wild types for either strong or weak eyes (*Figure 7D*), although variability was higher for *cdh2ΔEC* larvae. We also observed a significant decrease in the eye range of each eye in *cdh2ΔEC* larvae (*Figure 7E*), with a more pronounced effect observed for the strong

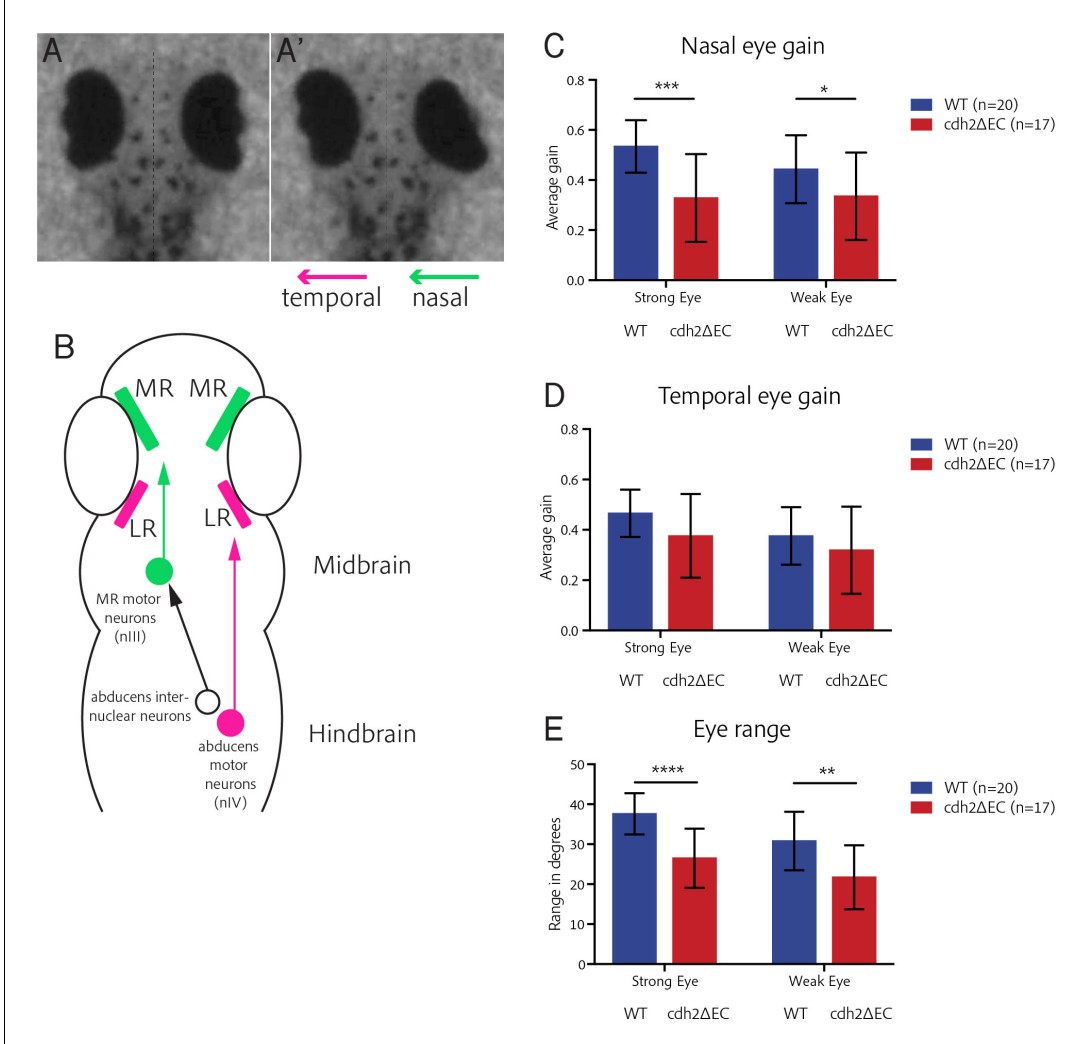

**Figure 7.** *cdh2ΔEC* larvae display impaired efficiency of nasal eye movements and defects in eye range. (**A–A'**) Example OKR video still illustrating coordinated nasal (right eye) and temporal (left eye) eye movements. (**B**) Ocular motor circuitry of underlying the OKR of larval zebrafish. Nasal and temporal eye movements are driven by medial and lateral rectus-innervating motor neurons of distinct nuclei. (**C–D**) Average nasal (**C**) and temporal (**D**) eye gain of wild types and *cdh2ΔEC* larvae averaged for all trials. There is a significant difference in nasal eye gain for the strong (p=0.0003) and weak eye (p=0.0472). There was no significant difference in temporal eye gain for the strong (p=0.0597) or weak eye (p=0.2574). (**E**) Total eye range for strong and weak eyes for each genotype. There was a significant difference for the strong (p<0.0001) and weak eye (p=0.0011). (**C–E**) All tests are t-test with Welch's corrections. Error bars are mean and SD. Both wild type and *cdh2ΔEC* larvae are from two clutches.

The online version of this article includes the following source data for figure 7:

**Source data 1.**

eye. Taken together, these data show that the efficiency of nasal eye movements (mediated by the MR muscle) as well as overall range of movement is significantly impaired in larvae in which cadherin-mediated adhesion in oculomotor neurons is reduced. Our finding that the temporal eye movements of the OKR are not impaired is consistent with a specific effect of *cdh2ΔEC* expression on MR-driven motor control rather than visual impairment and suggests that the clustered arrangement of these neurons is important for the function of the motor circuit.

## Discussion

In this study, we investigated the role of cadherin-mediated adhesion in the development of subnuclear topography, a unique phenomenon whereby clusters of neurons innervating functionally-related muscles via distinct nerve branches further group together, and its effect on motor function.

We demonstrate that cadherins control distinct aspects of cranial motor neuron positioning, including the clustering of neurons belonging to dorsal oculomotor subnuclei and the contralateral migration of ventral SR neurons, while also regulating subnuclear position through adhesive interactions between constituent oculomotor subnuclei. In addition, we find that eye movements driven by oculomotor neurons in which clustering is perturbed are defective, raising the possibility that their assembly into nuclei is important for circuit function.

## Cadherins play roles in the development of specific subnuclei

Our findings show that cadherin-based interactions likely coordinate the positioning of motor neurons both *within* and *between* subnuclei. This is evidenced by the observations that dorsal nIII (IR/MR) and ventral (SR) neurons display differential positioning defects when cadherin function is disrupted in these subnuclei, and that the location of the adjacent IO subnucleus is shifted following this perturbation. It has previously been proposed that cadherins globally expressed by motor neurons provide a baseline level of adhesion, while selective expression of other cadherins facilitates their sorting into distinct nuclei (*Demireva et al., 2011*). We find that several cadherins, including *cdh2* and *cdh11*, are expressed by all subnuclei, highlighting them as prime candidates to drive baseline levels of adhesion via homophilic interactions. Recent work raises the possibility that heterophilic interactions between type II cadherins belonging to specificity groups, such as *cdh8* (expressed to varying degrees by IR/MR and SR) and *cdh11* (expressed by all ocular motor neurons), could also participate, in collaboration with *cdh2* (*Brasch et al., 2018*; *Dewitz et al., 2019*).

Although we find that that the clustering of dorsal nIII neurons requires cadherin adhesivity, as previously shown for other cranial motor nuclei (*Astick et al., 2014*), ventral SR neurons do not appear to depend on cadherin function in the same way – here cadherins regulate transmedial migration. Dorsal nIII neurons, which are generated earlier and migrate dorsally, express additional cadherins during development. As such, their coalescence may be more cadherin-dependent. In addition, dorsal subnuclei differentially express several classical cadherins, including *cdh4* and *cdh10a*. In the future, defining the unique transcriptional identities of individual nIII subnuclei would allow us to manipulate cadherin expression profiles will allow us to uncover the specific roles of these cadherins.

## Contralateral migration of SR neurons

SR-innervating oculomotor neurons are one of a small number of neuronal cell types known to complete transmedial migration during development (*Simon and Lumsden, 1993*; *Taniguchi et al., 2002*). We identify cadherins as one of the few molecular factors regulating this distinctive migration. Our analyses show that they facilitate the timely convergence of SR neurons towards the midline and are required for their subsequent ability to cross it, suggesting that contralateral migration occurs in two steps. Cadherins are known to contribute to cell migration by coupling to retrograde actin flow, thereby acting as a molecular clutch (*Bard et al., 2008*; *Garcia et al., 2015*). This constitutes one possible mechanism by which cadherins facilitate the movement of SR neurons across the midline; whether the migration is driven by multiple cadherins or a single cadherin such as *cdh2*, such as in the case of tangential migration of facial branchiomotor neurons, is unknown (*Wanner and Prince, 2013*; *Rebman et al., 2016*).

SR migration is underlain by tight spatial and temporal control gated by Slit/Robo-mediated repulsion from the midline (*Bjorke et al., 2016*), while midbrain dopaminergic neurons have been proposed as a possible source of chemoattractants (*Puelles, 1978*). Expression of Neuropilins and their class III Semaphorin ligands is dynamic and differential during nIII development, suggesting that the timing of SR migration depends on these cues and receptors (*Chilton and Guthrie, 2003*). While exact roles are yet to be established experimentally, the dynamic expression of receptors could mean that they are downregulated at the time at which migration is usually complete. Delayed convergence resulting from perturbed cadherin-mediated adhesion could therefore affect the ability of SR neurons to respond to guidance cues at the appropriate time. What is the importance of SR contralateral migration for circuit function? The organisation of ocular motor circuitry is highly stereotyped, whereby motor neuron groups controlling coactive muscle pairs on opposite sides of the body are found on the same side of the brain. The failure of SR neurons to complete contralateral migration is likely to be detrimental to motor function, as contralateral innervation of the SR muscle

is key to the vertical eye movements of the gaze-stabilising vestibulo-ocular reflex (VOR; *Schoppik et al., 2017*).

## Role of cadherins in the development of motor circuit function

Our data show that although the nasal OKR is significantly perturbed when cadherin-mediated adhesion is reduced, the peripheral connection between motor neuron and muscle is left intact, suggesting that these deficits arise as a result of impaired central connectivity of dorsal MR neurons, which fail to coalesce. How might the stereotyped positioning of MR neurons driven by cadherins serve circuit function? MR neurons receive inputs from abducens internuclear neurons (*Cabrera et al., 1992*), which coordinate conjugate eye movements and could plausibly rely on correct MR neuron positioning for targeting. Whereas IR and SR neurons are innervated by central vestibular neurons, which in lateral-eyed vertebrates drive the torsional and vertical eye movements of the VOR (*Bianco et al., 2012*; *Schoppik et al., 2017*), these neurons may utilise a similar mechanism to connect to their oculomotor neuron partners. It would therefore be interesting to test whether the mispositioning of dorsal oculomotor neurons observed as a result of cadherin perturbation leads to a failure of presynaptic populations to correctly innervate their targets. Equally, if SR neurons which have failed to migrate across the midline were now innervated by central vestibular neurons that would usually synapse onto their contralateral counterparts, one might expect defects in the vertical component of the VOR. Together, these experiments would provide compelling evidence that spatial positioning is a major developmental strategy enabling oculomotor neurons to acquire the correct afferent input.

Indeed, previous studies indicate that somal positioning is crucial for arriving sensory afferents: in the mammalian spinal cord, clustering and correct positioning of motor neurons along the dorsoventral axis ensures input specificity and correct motor functioning; similar organisational principles also extend to premotor interneurons which segregate along the mediolateral axis according to their antagonistic output (*Sürmeli et al., 2011*; *Tripodi et al., 2011*). In contrast, respiratory behaviours driven by zebrafish facial branchiomotor neurons appear to be robust to manipulations that shift their rostrocaudal location but do not otherwise alter their subnuclear topography, suggesting that they still receive inputs despite perturbed positioning (*McArthur and Fetcho, 2017*). As such, some aspects of positioning may be more important for the acquisition of input than others.

In addition to controlling somal location, one further possible mechanism to mediate the effect of cadherins on larval eye movements could relate to findings that cadherin-dependent clustering of motor neurons underlies the development of gap junction coupling and spontaneous activity (*Montague et al., 2017*), both of which are known to contribute to mature activity patterns required for motor control (*Warp et al., 2012*; *Chang et al., 1999*; *Korn and Bennett, 1975*). Cadherins may also contribute to selective molecular recognition between neurons within sensorimotor circuits, indeed type II cadherin expression has been shown to delineate functional circuits and control diverse cellular interactions throughout the CNS (reviewed in *Redies, 2000* and *Basu et al., 2015*; *Friedman et al., 2015*; *Duan et al., 2018*). Circuit function may therefore depend on classical cadherins both for the coordination of cell body positioning, placing critical spatial constraints on connectivity between pre- and postsynaptic partners, and for specificity at the level of central synapses between motor neurons and their inputs.

## Importance of subnuclear topography

Our data show that cadherin-based interactions between subnuclei control the global positioning of a single oculomotor subnucleus, in addition to those which drive the precise positioning of neurons within individual subnuclei. What is the advantage of subnuclear topography? In the mammalian spinal cord, the layout of somatic motor neurons adheres to a higher order spatial plan, whereby pools innervating functionally related limb muscles are further grouped into columns along the dorsoventral axis (*Romanes, 1964*; *Vanderhorst and Holstege, 1997*). An analogous organisational hierarchy may be present in the brainstem, where some cranial motor nuclei are composed of adherent subnuclei which innervate muscles with synergistic functions. With the exception of the MR subnucleus, which receives input from abducens internuclear neurons, oculomotor subnuclei innervating related muscle pairs receive inputs from the same population of central vestibular neurons, which participate either in the torsional upward (IO-SR) or downward (SO-IR) eye movements of the VOR (*Bianco et al., 2012*; *Schoppik et al., 2017*). The grouping of these subnuclei may therefore

represent an efficient developmental strategy by which functionally related neuronal populations acquire inputs from interneurons which drive reflex circuits. The multi-layered organisation of motor neurons is therefore likely to have important implications for the development of connectivity. During nucleogenesis, classical cadherins are likely to co-participate with several other cell adhesive systems to give rise to the stereotyped anatomical arrangement of ocular motor subnuclei. Other studies have highlighted roles for protocadherins, as well as nectin-afadin signalling in the positional control of developing motor neurons (*Dewitz et al., 2018*; *Asakawa and Kawakami, 2018*). Our analyses reveal the complexity of motor neuron positioning programmes regulated by a single molecular family within a functional motor circuit. The precise anatomy, robust function, and tractability of the zebrafish ocular motor system make it uniquely suited for further investigations into the structure-function relationships underlying circuit formation.

# Materials and methods

## Key resources table

| Reagent type (species) or resource | Designation | Source or reference | Identifiers | Additional information |
|---|---|---|---|---|
| Genetic reagent (*Danio rerio*) | *Tg(Isl1:GFP)* | *Higashijima et al., 2000* | RRID:ZFIN_ZDB-GENO-070810-2 | |
| Genetic reagent (*Danio rerio*) | *Tg(isl1:cdh2ΔEC-mCherry)vc25/Tg(Isl1:GFP)* | *Rebman et al., 2016* | | |
| Antibody | anti-GFP (chicken polyclonal) | Abcam | RRID:AB_300798 | 1:500 dilution |
| Antibody | anti-RFP (rabbit polyclonal) | MBL Life Science | RRID:AB_591279 | 1:200 dilution |
| Antibody | Anti-acetylated tubulin (rabbit monoclonal) | Cell Signalling Technologies | RRID:AB_10544694 | 1:1000 dilution |
| Antibody | Anti-myosin heavy chain (mouse monoclonal) | Developmental Studies Hybridoma Bank (DSHB), University of Iowa | RRID:AB_528356 | 1:200 dilution |
| Antibody | Anti-SV2A (mouse monoclonal) | Developmental Studies Hybridoma Bank (DSHB), University of Iowa | RRID:AB_2315387 | 1:200 dilution |
| Chemical compound, drug | Tetramethylrhodamine-α-bungarotoxin conjugate | Invitrogen | RRID:AB_2313931 | 1:1000 dilution |
| Commercial assay, kit | RNAscope Fluorescent Multiplex Reagent kit | ACD Bio | Cat# 320850 | |
| Software, algorithm | FIJI/Image J | https://github.com/fiji/fiji | RRID:SCR_002285 | |
| Software, algorithm | Imaris 9 | Bitplane | RRID:SCR_007370 | |
| Software, algorithm | MATLAB | Mathworks | RRID:SCR_001622 | |
| Software, algorithm | JAGS | https://cran.r-project.org/web/packages/rjags/index.html | RRID:SCR_017573 | |
| Software, algorithm | LabView | National Instruments | RRID:SCR_014325 | |
| Software, algorithm | Prism7 | GraphPad | RRID:SCR_002798 | |

## Experimental model and transgenic lines

Zebrafish were maintained at 28.5°C on a 14/10 hr light/dark cycle. Embryos were collected after timed matings and raised in Danieau solution. Staging was performed according to hours post-fertilisation (hpf) as previously described (*Kimmel et al., 1995*). With the exception of behavioural experiments, 0.003% phenylthiourea was added at 24 hpf in order to prevent pigment formation. Wild-type, *Tg(Isl1:GFP)* (*Higashijima et al., 2000*; ZFIN Cat# ZDB-GENO-070810–2, RRID:ZFIN_ZDB-

GENO-070810-2) and homozygous *Tg(isl1:cdh2ΔEC-mCherry)vc25/Tg(Isl1:GFP)* (*Rebman et al., 2016*) zebrafish strains were used in the present study. The latest developmental time point examined was at 5 days post-fertilisation (dpf), at which time the sex of individual larvae has not yet been determined. All procedures received Home Office approval and were reviewed by the local Ethics Committee.

## Immunohistochemistry

Antibody labeling was performed on whole-mount embryos as previously described (*Hunter et al., 2011*). Following fixation, embryos were treated with 0.25% trypsin in phosphate-buffered saline (PBS) for 2 min at 50 hpf and 3 min at 72 hpf and 4 dpf. To label ocular motor neurons at 50 hpf, the CNS was dissected. For synaptic component labelling, larvae were fixed in 4% paraformaldehyde (PFA) for 2 hr and permeabilised using 0.25% trypsin in PBS for 25 mins at room temperature (RT). Larvae were then bisected and incubated in primary antibody solution in PBS-Triton 2% and 1% dimethyl sulfoxide (DMSO) for 2 days at 4°C. The following antibodies were used: anti-GFP 1:500 (Abcam; RRID:AB_300798), anti-RFP 1:200 (MBL Life Science; RRID:AB_591279); anti-acetylated tubulin 1:1000 (Cell Signalling Technologies; RRID:AB_10544694); anti-myosin heavy chain 1:200 (Developmental Studies Hybridoma Bank (DSHB), University of Iowa; RRID:AB_528356); anti-SV2A 1:200 (DSHB; RRID:AB_2315387). Acetylcholine receptors of post-synaptic compartments were stained using a tetramethylrhodamine-α-bungarotoxin conjugate 1:1000 (Invitrogen; RRID:AB_2313931).

## RNAscope in situ hybridisation

For detection of mRNA species, RNAscope in situ hybridisation was performed as previously described using an RNAscope Fluorescent Multiplex Reagent kit (320850; ACD, Hayward, CA, USA) using custom-designed probes, with some modifications made to the length of fixation and probe hybridisation and the washing reagent used, as follows (*Gross-Thebing et al., 2014*): Whole embryos were fixed for 6 hr at RT in 4% PFA. For larvae fixed at 72 hpf, the CNS was dissected following fixation. 1x RNAscope wash buffer (310091; ACD, Hayward, CA, USA) in ddH$_2$0 was used for all wash steps. Probe hybridisation was performed at 45°C for 22 hr to ensure full tissue penetration. The following probes were tested: *cdh2, cdh4, cdh7a, cdh8, cdh10a, cdh11, cdh13*, as well as *foxg1a* as a positive control and *dapB* as a negative control. Probe detection was performed using ATTO 550 fluorescent dye and the same imaging parameters were maintained between specimens.

## Retro-orbital dye fills

For DiI fills, embryos or larvae were fixed at relevant developmental stages in 4% PFA at 2 hr at RT, or overnight at 4°C. Specimens were then carefully placed in a small petri dish with a sylgard-coated bottom containing PBS and secured with the eye facing up using a Tungsten minuten pin. The eye was carefully removed from the orbit with sharp forceps and DiI (D3911, Invitrogen) was applied to the tissue at the corner of the orbit, where the oculomotor and trochlear nerves enter, using a blunt microinjection needle held by a micromanipulator. DiI was left to bind to the tissue for 3–5 min, after which embryos were stored overnight in PBS at 4°C. The following day, brains were dissected from whole larvae and mounted for imaging. For live retro-orbital dye fills, ocular motor nuclei were retrogradely labelled as previously described (*Greaney et al., 2017*). Crystallised fluorescently-conjugated Alexa Fluor 647 dye (10,000 molecular weight, Thermo Fisher D-22914) was applied unilaterally to the orbit of anesthetised fish at four dpf and the side of incision and dye application was recorded for each larva. Larvae were returned to the incubator at 28.5°C to recover overnight before imaging at 5 dpf.

## Identification of retro-orbital dye-filled neurons as specific ocular motor neurons

In wild-type larvae, dye-filled neurons can be assigned to specific extraocular muscles (EOMs) based on the lack of GFP expression in the IO-innervating population in the *Tg(Isl1:GFP)* line (*Higashijima et al., 2000*; *Greaney et al., 2017*), as well as known projection patterns of ocular motor neurons across vertebrates (*Evinger, 1988*), whereby superior EOMs are innervated by contralateral ocular motor neurons. The SO-innervating nIV is visible as a discrete nucleus separated

from nIII by the midbrain-hindbrain boundary (MHB), therefore contralateral dye-filled nIV neurons are easily distinguished from nIII neurons. Contralateral GFP-positive neurons belonging to nIII are therefore classified as SR neurons. Dye-filled ipsilateral neurons are therefore assigned based on presence or absence of GFP fluorescence: GFP-negative neurons belong to the IO subnucleus, whereas GFP-positive neurons are IR/MR neurons.

## Microscopy and image analysis

For all in vivo and ex vivo preparations, embryos were mounted in 1% low-melting point (LMP) agarose and imaging was performed on a Zeiss LSM880 Laser Scanning confocal microscope with a 20x (NA 0.95) water-immersion objective. All images were acquired as z-stacks with optical sections spaced at 1 µm. For image analysis, Fiji/Image J (RRID:SCR_002285) and Imaris 9 (Bitplane; RRID: SCR_007370) software was used.

## Analysis of cadherin expression using RNAscope in situ hybridisation

For quantification of RNAscope in situ hybridisation fluorescent signal of *Isl1:GFP* embryos/larvae, the GFP signal was thresholded to select nIII and nIV using the Imaris Absolute Intensity thresholding function and subsequently used to obtain a masked ATTO 550 channel. For quantification of ATTO 550 fluorescence within nIV and the three defined anatomical regions of nIII, five equally spaced slices along the dorsoventral (DV) axis were selected from each substack (excluding the first and last slice). Regions of interest (ROIs) were selected using the GFP signal and the mean grey value of ATTO 550 fluorescence was measured within ROIs for each slice. Average background fluorescence was computed for each specimen by measuring and subsequently averaging the mean grey value of ATTO 550 fluorescence within three GFP-positive ROIs within sections of the ocular motor nerves that did not contain visible ATTO 550-positive puncta. Measurements for each slice were performed on both sides on either side of the midline and corrected by subtracting the average background mean grey value, prior to averaging within each specimen. Each probe was quantified in three independent embryos/larvae.

## Analysis of dorsal oculomotor neuron clustering

Dorsal oculomotor neurons were defined as neurons located in the dorsal half of the oculomotor nucleus as ascertained by finding the first and last optical section in the z-stack. Neurons were recorded using the Imaris 9 (Bitplane; RRID:SCR_007370) Spots Detection tool using parameters optimised for detection of three-dimensional centre coordinates ~6 µm across and manual curation. Coordinates were measured for each nucleus on either side of the midline for each larva. Measures derived from these coordinates were averaged within each larva to provide one value for each specimen. Delaunay tessellations were applied to the coordinates using a custom written MATLAB script (RRID:SCR_001622) to provide estimates of local cell densities, code available on GitHub (https://github.com/aknufer/MN_delaunay; *Knüfer, 2020*, copy archived at swh:1:rev: 471db6d5a903f3a8aaa4de98f5a0b1b63968b6b0).

## Image analysis of dye-filled neurons

In order to prevent bleed-through of Alexa 647 fluorescent signal into the 488 nm channel, control dye-filled larvae without GFP fluorescence and GFP-positive larvae without dye-fills were imaged in order to establish appropriate imaging parameters. Motor neuron identity was assigned using simultaneous visualisation of confocal z-stacks in Imaris 9 (Bitplane; RRID:SCR_007370) and Fiji/Image J (RRID:SCR_002285) and cells were only counted which revealed a smooth distribution of dextran labelling. Cell locations were recorded using the Cell Counter functionality in Fiji/Image J (Kurt de Vos, University of Sheffield) while 3D-visualisation in Imaris 9 (Bitplane; RRID:SCR_007370) was employed to distinguish between GFP-positive and -negative cells. The use of interactive 3D-visualisation in Imaris 9 offered considerable advantages to 2D-visualisation when distinguishing dimly fluorescent GFP-positive, dye-filled neurons from GFP-negative, dye-filled neurons. Adjustments to brightness and contrast using the Imaris Display Adjustment function revealed that the former possessed cell-shaped GFP signal which could be visualised from different angles, while the latter had no or only unstructured and noisy contaminating GFP signal from adjacent neurons.

## Analysis of spatial distribution of IO cells

Confocal z-stacks were split into 15 and 4 equally spaced dorsoventral (DV) and mediolateral (ML) subdivisions (*z0* to *z14*, ventral to dorsal; *x0* to *x3*, medial to lateral) per larva. This was based on the first and last GFP- or Alx647-positive nIII cell for DV subdivisions and on the distance between the lateral most nIII cell and the midline for ML subdivisions. Each ipsilateral GFP-negative dye-filled neuron was assigned a DV and ML coordinate based on location. We adopted a Bayesian approach to provide a non-parametric characterisation of the spatial distribution of IO neurons. We attributed an occupancy probability to each of the $K = 15 \times 4$ mutually exclusive bins across DV and ML axes. Occupancy probabilities were modelled by a Dirichlet distribution with corresponding parameters estimated from the observed data using statistical inference. We used custom R scripts employing the JAGS library (RRID:SCR_017573) to draw 20,000 Markov Chain Monte Carlo (MCMC) samples from the posterior distribution of the occupation probabilities, code available on GitHub (https://github.com/giovannidiana/MN_topo). This method allowed us to estimate average occupancy distributions as well as confidence intervals. To examine the overall difference between the posterior distributions of IO neuron location between genotypes, we estimated the Kullback-Leibler (KL) divergence between their occupancy distributions across the MCMC samples. To choose a sufficient number of posterior samples we used the stability of the posterior *KL* distribution as convergence criterion. The directionality of shifts in IO location between genotypes along DV (*z-shift*) and ML (*x-shift*) axes was computed by calculating the difference between the average DV and ML positions from the occupancy probabilities of each genotype (e.g. *z_shift = z_shift_wt – z_shift_cdh2ΔEC*).

## Analysis of axon pathways and neuromuscular junctions

For analysis of ocular motor axons, FIJI Cell Counter plugin was used to manually count numbers of axon branches for each target muscle in confocal z-stacks. Branches were counted within a 50 μm x 50 μm window. All branches were counted if they (a) innervated a target muscle or (b) emerged as secondary processes from a branch innervating a target muscle. Branch width was not considered. For analysis of NMJs, synaptic sites were identified based on their anatomical location in the orbit using brightfield signal. Colocalisation of pre-synaptic and post-synaptic components was performed on ROIs of substacks taken from larger confocal z-stacks.

## Optokinetic reflex (OKR) assay and analysis

Horizontal eye movements were assayed at five dpf using a custom-built apparatus similar in design to *Brockerhoff, 2006* and controlled using custom LabView (National Instruments, RRID:SCR_014325) routines. Larvae were mounted singly in a small drop of 2% LMP agarose (A9414, Sigma) with the dorsal side up, in the middle of a 35 mm non-treated petri dish, to which 1x Danieau solution was added. For OKR recordings, we used a projector (AAXA P2 Jr) and a cold mirror (Thorlabs) to project vertical stripes onto a cylindrical screen. Larvae were positioned in the centre of this screen. A 850 nm LED array illuminated the fish from below. Imaging was performed at 100 Hz using a camera (Thorlabs) fitted with a machine vision lens and a filter (R72 52 mm HOYA) to block visible light. Each larva was presented with a horizontally moving vertical grating of varying stimulus parameters, including varying speeds, spatial frequency and contrast using Psychophysics Toolbox (*Brainard, 1997*). Gratings reversed direction every 6 s. Eye position was tracked online by thresholding images and computing the central moments of the two largest binary objects, which invariably corresponded to the eyes. The gain of OKR slow-phase movements was quantified separately for left and right eyes as well as for nasal and temporal directions, as the ratio of eye velocity to grating velocity (custom MATLAB scripts). The range of eye movement (from the resting position of the eye towards rostral or caudal positions in degrees) was evaluated as the maximum/minimum position for each eye. Strong and weak eyes for each larva were assigned based on the eye displaying the greatest eye range.

## Quantification and statistical analysis

All standard statistical analyses were performed in Prism7 (Graphpad; RRID:SCR_002798) or MATLAB (RRID:SCR_00162). For normally distributed data, parametric tests were used to make statistical comparisons between groups. For one or more independent groups, Welch's t-tests (with the Holm-Sidak method for multiple comparisons, where applicable) were used. For non-normally distributed

data, non-parametric tests were used to make statistical comparisons between groups. For two independent groups, the Mann-Whitney U test was used. For multidimensional data, a multivariate extension of the Kolmogorov-Smirnov test was applied. In order to assess differences between experimental conditions with categorical outcomes, chi-squared-tests were used. Bayesian statistics were applied to perform non-parametric estimation of the spatial organisation of IO neurons. Wherever possible, analyses were performed blind. For behavioural experiments, the number of experimental animals per group was chosen to approximate or slightly exceed that of previously published studies employing a similar experimental design (*Schoonheim et al., 2010*; *Bianco et al., 2012*).

## Acknowledgements

We thank Corinne Houart for giving us the *foxg1a* probe for in situ hybridisation, David Schoppik for assistance with the retro-orbital dye fill technique, Isaac Bianco and Ivana Poparic for initial assistance with the OKR assay and Stephen Price for valuable advice and discussion. This work was supported by a Biotechnology and Biological Sciences Research Council PhD studentship and The Company of Biologists Travelling Fellowship to AK, a Medical Research Council grant to SG and a Wellcome Investigator Award to JC.

## Additional information

### Funding

| Funder | Grant reference number | Author |
|---|---|---|
| Biotechnology and Biological Sciences Research Council | BB/J014567/1 | Athene Knüfer |
| Company of Biologists | DEV-170218 | Athene Knüfer |
| Wellcome Trust | 102895/Z/13/Z | Jonathan DW Clarke |
| Medical Research Council | MR/L020742/2 | Sarah Guthrie |

The funders had no role in study design, data collection and interpretation, or the decision to submit the work for publication.

### Author contributions

Athene Knüfer, Conceptualization, Software, Formal analysis, Investigation, Methodology, Writing - original draft, Writing - review and editing; Giovanni Diana, Software, Formal analysis, Writing - review and editing; Gregory S Walsh, Resources, Writing - review and editing; Jonathan DW Clarke, Sarah Guthrie, Conceptualization, Formal analysis, Investigation, Methodology, Writing - original draft, Writing - review and editing

### Author ORCIDs

Athene Knüfer https://orcid.org/0000-0003-4321-4239
Giovanni Diana http://orcid.org/0000-0001-7497-5271
Sarah Guthrie https://orcid.org/0000-0002-8446-9150

### Ethics

Animal experimentation: This work was approved by the local Animal Care and Use Committee (King's College London) and was carried out in accordance with the Animals (Experimental Procedures) Act, 1986, under licence from the United Kingdom Home Office (PPLs: 70/7753 and P70880F4C-Z001, PIL: I1D87502D).

### Decision letter and Author response

Decision letter https://doi.org/10.7554/eLife.56725.sa1
Author response https://doi.org/10.7554/eLife.56725.sa2

## Additional files

**Supplementary files**
- Transparent reporting form

### Data availability

The data that support the findings in this study are available within the article and supporting files.

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
