## [Decision Letter]

**Acceptance summary:**

In this nicely written and conducted study, Guthrie and colleagues characterize the spatial patterns of cadherin expression in select oculomotor neurons, then use targeted expression of a dominant negative cadherin to study the role of cadherins in neuronal positioning and circuit function. Their results demonstrate a role for cadherins in aspects of oculomotor neuron positioning, supporting an important role for cadherins in normal oculomotor circuit development. The authors capitalize on the unique strengths of their model system to make an important contribution to mechanistic and functional understanding of neuronal positioning in hindbrain.

**Decision letter after peer review:**

Thank you for submitting your article "Cadherins regulate nuclear topography and function of developing ocular motor circuitry" for consideration by *eLife*. Your article has been reviewed by three peer reviewers, and the evaluation has been overseen by Marianne Bronner as the Senior and Reviewing Editor. The following individual involved in review of your submission has agreed to reveal their identity: Kimberly McArthur (Reviewer #1).

The reviewers have discussed the reviews with one another and the Reviewing Editor has drafted this decision to help you prepare a revised submission.

Summary:

This nicely written and generally well-presented study uses the zebrafish model's advantages to integrate an investigation of the molecular basis of nucleogenesis with behavioral output. The authors interrogate the role of cadherins in the oculomotor system, cleverly combining expression analysis with functional modulation using a dominant-negative transgenic tool. Careful imaging, quantitation, cell-tracing, and immunolabeling reveal key changes in the distributions of neurons, but not their connectivity.

Essential revisions:

1) The lack of quantitative analysis of Figure 1 (cadherin expression) stands in contrast to the rest of the paper, and should be improved.

2) The paper would be significantly strengthened by the addition of data on how the scattered neurons are or are not receiving normal presynaptic innervation. The functional analysis of eye movement doesn't show that the behavioral defects are because of the neuron mis-positioning. A specific prediction (since muscle innervation is not affected) is that presynaptic neurons fail to innervate these mispositioned motor neuron appropriately. The vestibular neurons that synapse on nIII and nIV motor neurons have been described (PMID:28972121). While we feel this would be a nice addition, we understand that it may not be feasible at the present time, so this should be viewed as an opportunity to add data or at least a discussion of this point but additional physiological data are not a requirement.

Reviewer #1:

In this study, the authors characterize spatial patterns of cadherin expression in select oculomotor neurons, then use targeted expression of a dominant negative cadherin to study the role of cadherins in neuronal positioning and circuit function. Their results demonstrate a role for cadherins in some aspects of oculomotor neuron positioning, and strongly suggest a role for cadherins in normal oculomotor circuit development. Knüfer and colleagues capitalize on the unique strengths of their model system to make an important contribution to our mechanistic and functional understanding of neuronal positioning in hindbrain -- building on previous work, and illuminating new avenues forward. Further, in my opinion, this work is presented with clarity, rigor, and integrity. I highly recommend this paper for publication in *eLife*.

Reviewer #2:

An important capstone to the study, which also renders it of significance appropriate to *ELife*, is the functional analysis of the optokinetic reflex, which the authors show to depend upon proper nuclear organization.

1) The Discussion in general does a nice job of putting the work into a broad context. However, I find the last paragraph of the subsection “Role of cadherins in development of motor circuit function”, a little confusing: what exactly is the last comment about 'specificity at the level of individual synapses' referring to? Given that the data provided in the manuscript do not suggest changes in synaptogenesis, I wonder if the authors are thinking about possible defects in Abducens connections? I think this section just needs a little fleshing out.

2) While the quality of the data panels is in general very high, there are several places where the clarity of the figures could nevertheless be significantly improved, more details are laid out below, but overall it is important to ensure optimal clarity for the reader.

Reviewer #3:

This manuscript focuses on the process of nucleogenesis, whereby groups of neurons that innervate the same targets cluster together within the central nervous system. Sometimes, motor neurons form sub-nuclei that innervate distinct the muscle targets of a motor single nerve. This paper focuses on the oculomotor neurons which form three distinct sub-clusters, asking: what the molecular mechanism of sub-cluster formation? The authors start by describing the expression of a number of type-I and type-II cadherins in the oculomotor and nearby trochlear nuclei. Then they use isl1 to drive dominant-negative form of *Cdh2* to block all cadherin adhesive function in motor neurons in a stable transgenic line and describe the consequences on oculomotor sub-nuclear organization. They observe a looser, scattered organization and a failure of DN-expressing neurons to complete a characteristic cross-midline migration, which they detect using retrograde labeling. They also observe a subtle mispositioning of DN-expressing trochlear neurons, and even more subtle non-autonomous effect on the positions of inferior-oblique innervating neurons. Finally, they show a specific defect in the eye movements driven by oculomotor neurons in an optokinetic response assay.

While the findings are clearly presented and meticulously quantified, the use of a single, blunt instrument (*cdh2ΔEC*) is insufficient to understand the function of cadherin-based adhesion in nucleogenesis. That cadherin-based adhesion is important for motor neuron segregation into nuclei is not unexpected, since these and other authors have shown it to be the case for other cranial and spinal motor nuclei. The more mechanistic questions-about which cadherins cause the segregation of which motor neurons from each other and/or from other cells in their environment, and at what stage of motor neuron development this happens-are not addressed using this one tool. Consequently, I feel that the paper does not constitute enough of an advance to be appropriate for publication in *ELife*.

Figure 1: The authors schematize cadherin expression in a pseudo-quantitative format in Figure 1B, C. Despite their skillful quantification of phenotypes in the rest of the paper, this map appears to be qualitative, and does not always appear to match the representative figures. Furthermore, not all representative images used to generate this map are shown. For instance, transverse images are not shown for the genes analyzed in Figure 1—figure supplement 1 and Figure 1—figure supplement 2. These should be included. For instance, for *cdh4* the authors indicate different expression levels between the ventral/medial and dorsal/medial domains, and similar expression levels between the dorsal/medial and dorsal/lateral domains, neither of which appear consistent with the image in Figure 1I. This map could be improved by quantitative measurement of fluorescence levels in the three domains in their 3D images across multiple animals. If the font sizes in the schematic are based on pixel intensities in the various motor nucleus regions, how many embryos were used? In some cases, expression is not bilaterally symmetric, for example in the dorsal view of *cdh10a*. Is this asymmetry consistent from embryo to embryo?

Figure 1: the evaluation of cadherin expression is done at 48 hpf, after motor innervation of eye muscles is complete and oculomotor neurons are at least partly subdivided into subnuclei. What was cadherin expression like when the motor neurons are first detectable by isl1:GFP expression? Presumably it is from this earlier stage that *cdh2ΔEC* will disrupt the process of sub-nucleogenesis.

Figure 5: In the Results and the Discussion, the authors frame the non-autonomous effect of *cdh2ΔEC* expression on IO neurons as evidence of cadherin-based interactions between different sub-nuclei. While this is one possible interpretation, it is equally likely that changes in the organization of the other subnuclei (e.g. shifts in the position, span, or density of these subnuclei) could cause subtle shifts in the IO nucleus position indirectly by mechanical means. For instance, expanding the volume taken up of the dorsal subnucleus could push the IO nucleus more ventrally. If the authors could show that *cdh2ΔEC* expression in IO neurons and not other nIII neurons had the same effect on both populations, their conclusion of cadherin-based interactions between them would be better supported. As it stands, the authors should consider both direct and indirect mechanisms.

---

## [Author Response]

Essential revisions:1) The lack of quantitative analysis of Figure 1 (cadherin expression) stands in contrast to the rest of the paper, and should be improved.

This is an excellent suggestion. We have now added quantitative analysis of cadherin expression to Figure 1 and its associated supplements, which supports our original findings that cadherin expression is dynamic and differential within developing ocular motor neurons, but has provided further nuance to our analysis. We have also made changes to the expression profile schematics in Figure 1 and Figure 1—figure supplement 3 in order to reflect varying levels of fluorescence intensities within ocular motor domains. Further details are given in response to reviewer 3’s comments.

2) The paper would be significantly strengthened by the addition of data on how the scattered neurons are or are not receiving normal presynaptic innervation. The functional analysis of eye movement doesn't show that the behavioral defects are because of the neuron mis-positioning. A specific prediction (since muscle innervation is not affected) is that presynaptic neurons fail to innervate these mispositioned motor neuron appropriately. The vestibular neurons that synapse on nIII and nIV motor neurons have been described (PMID:28972121). While we feel this would be a nice addition, we understand that it may not be feasible at the present time, so this should be viewed as an opportunity to add data or at least a discussion of this point but additional physiological data are not a requirement.

This is an excellent point and one that we had indeed previously considered. We have previously gathered data showing that neurons contained within an unpublished transgenic line (*Tg(Gad2:GAL4;UAS:TagRFP-CAAX)/Tg(Isl1:GFP)*) generated within Jon Clarke’s laboratory are putative inhibitory counterparts of the vestibular neurons described above and show contacts onto wild-type ocular motor neurons. However, we were unable to address this question with this line in our initial attempts due to its very weak expression and spectral overlap with the mCherry tag of the dominant negative cadherin construct. When Covid restrictions allow, it is our intention to undertake further experiments to examine the innervation of ocular motor neurons by its presynaptic neurons, which we will be happy to provide as a preprint to bioRxiv or submit as a Research Advance to *eLife*, and we have expanded our Discussion to reflect this.

Reviewer #2:[…] 1) The Discussion in general does a nice job of putting the work into a broad context. However, I find the last paragraph of the subsection “Role of cadherins in development of motor circuit function”, a little confusing: what exactly is the last comment about 'specificity at the level of individual synapses' referring to? Given that the data provided in the manuscript do not suggest changes in synaptogenesis, I wonder if the authors are thinking about possible defects in Abducens connections? I think this section just needs a little fleshing out.

This sentence has been amended to clarify that we speculate on synaptic connections between oculomotor neurons and their presynaptic inputs, rather than synapses between motor neurons and muscle.

“Circuit function may therefore depend on classical cadherins both for the coordination of cell body positioning, placing critical spatial constraints on connectivity between pre- and postsynaptic partners, and for specificity at the level of central synapses between motor neurons and their inputs.”

2) While the quality of the data panels is in general very high, there are several places where the clarity of the figures could nevertheless be significantly improved, more details are laid out below, but overall it is important to ensure optimal clarity for the reader.Reviewer #3:[…] While the findings are clearly presented and meticulously quantified, the use of a single, blunt instrument (cdh2ΔEC) is insufficient to understand the function of cadherin-based adhesion in nucleogenesis. That cadherin-based adhesion is important for motor neuron segregation into nuclei is not unexpected, since these and other authors have shown it to be the case for other cranial and spinal motor nuclei. The more mechanistic questions-about which cadherins cause the segregation of which motor neurons from each other and/or from other cells in their environment, and at what stage of motor neuron development this happens-are not addressed using this one tool. Consequently, I feel that the paper does not constitute enough of an advance to be appropriate for publication in eLife.Figure 1: The authors schematize cadherin expression in a pseudo-quantitative format in Figure 1B, C. Despite their skillful quantification of phenotypes in the rest of the paper, this map appears to be qualitative, and does not always appear to match the representative figures. Furthermore, not all representative images used to generate this map are shown. For instance, transverse images are not shown for the genes analyzed in Figure 1—figure supplement 1 and Figure 1—figure supplement 2. These should be included. For instance, for cdh4 the authors indicate different expression levels between the ventral/medial and dorsal/medial domains, and similar expression levels between the dorsal/medial and dorsal/lateral domains, neither of which appear consistent with the image in Figure 1I. This map could be improved by quantitative measurement of fluorescence levels in the three domains in their 3D images across multiple animals. If the font sizes in the schematic are based on pixel intensities in the various motor nucleus regions, how many embryos were used? In some cases, expression is not bilaterally symmetric, for example in the dorsal view of cdh10a. Is this asymmetry consistent from embryo to embryo?

These are excellent points and we have now added quantitative analysis of fluorescence intensities for all of our in situ hybridisation data in place of a qualitative assessment. This analysis has been performed for n=3 animals, for each time point and each cadherin within different regions of nIII, as well as nIV. We have updated our schematics in Figure 1 and Figure 1—figure supplement 3., Results and Materials and methods sections to reflect these analyses and have also included transverse images which were previously not shown for the genes analysed in Figure 1—figure supplement 1 and Figure 1—figure supplement 3 (previously Figure 1-supplement 2). In relation to Figure 1I, our new analyses confirm that *cdh4* expression is higher in dorsomedial nIII than in ventral nIII as identified before – as some cells anterior to the outlines shown of nIV show higher intensities than those seen most medially. However expression in dorsolateral nIII is indeed slightly higher than in dorsomedial nIII, as reviewer 3 has observed. On the whole, we find that expression levels are generally very similar on left and right sides, however in a few cases there may be small asymmetries such as that highlighted for *cdh10a* at 48 hpf. In our new analyses, we have averaged measurements for left and right sides for each slice analysed, within the same specimen.

Figure 1: the evaluation of cadherin expression is done at 48 hpf, after motor innervation of eye muscles is complete and oculomotor neurons are at least partly subdivided into subnuclei. What was cadherin expression like when the motor neurons are first detectable by isl1:GFP expression? Presumably it is from this earlier stage that cdh2ΔEC will disrupt the process of sub-nucleogenesis.

When deciding on an early time point, we chose to evaluate a time point at which substantial number of neurons have been born from each oculomotor population, despite nucleus organisation appearing less mature. As ventral neurons have been shown to be born later than dorsal neurons, with neurogenesis of this population completing ~50 hpf, we opted to assay cadherin expression at 48 hpf. Furthermore, our published data show that motor innervation of eye muscles in zebrafish is not complete until 72 hpf, and this was the rationale for choosing the later time point. We agree that oculomotor neurons would be expected to express cadherins as soon as they are postmitotic, however assaying this was beyond the scope of this study.

Figure 5: In the Results and the Discussion, the authors frame the non-autonomous effect of cdh2ΔEC expression on IO neurons as evidence of cadherin-based interactions between different sub-nuclei. While this is one possible interpretation, it is equally likely that changes in the organization of the other subnuclei (e.g. shifts in the position, span, or density of these subnuclei) could cause subtle shifts in the IO nucleus position indirectly by mechanical means. For instance, expanding the volume taken up of the dorsal subnucleus could push the IO nucleus more ventrally. If the authors could show that cdh2ΔEC expression in IO neurons and not other nIII neurons had the same effect on both populations, their conclusion of cadherin-based interactions between them would be better supported. As it stands, the authors should consider both direct and indirect mechanisms.

This is an excellent observation and an experiment that we would be eager to try, however there are currently no available tools to specifically target IO neurons, nor a defined transcriptional profile known for this population. Without this supporting evidence, we have therefore toned down our conclusion to include the possibility of indirect mechanisms as suggested.

“In sum, these results suggest that cadherin-dependent neuron-neuron adhesive interactions between oculomotor subnuclei also contribute to the appropriate positioning of the IO subnucleus. However we cannot rule out an alternative possibility that the observed shift in IO subnucleus position could also be a mechanically-driven consequence of changes to the arrangement of the other oculomotor subnuclei; novel tools to target expression of *cdh2ΔEC* to IO neurons specifically would allow us to address this possibility definitively.”